# TELL YOUR MODEL WHERE TO ATTEND: POST-HOC ATTENTION STEERING FOR LLMS

**Qingru Zhang**[†,∗]**, Chandan Singh**[◇]**, Liyuan Liu**[◇]**, Xiaodong Liu**[◇]**, Bin Yu**[‡]**,
Jianfeng Gao**[◇]**, Tuo Zhao**[†]

[†]Georgia Institute of Technology  [‡]University of California, Berkeley  [◇]Microsoft Research
`{qingru.zhang,tourzhao}@gatech.edu`
`binyu@berkeley.edu`
`{chansingh,lucliu,xiaodl,jfgao}@microsoft.com`

## ABSTRACT

In human-written articles, we often leverage the subtleties of text style, such as **bold** and *italics*, to guide the attention of readers. These textual emphases are vital for the readers to grasp the conveyed information. When interacting with large language models (LLMs), we have a similar need – steering the model to pay closer attention to user-specified information, e.g., an instruction. Existing methods, however, are constrained to process plain text and do not support such a mechanism. This motivates us to introduce PASTA – Post-hoc Attention STeering Approach, a method that allows LLMs to read text with user-specified emphasis marks. To this end, PASTA identifies a small subset of attention heads and applies precise attention reweighting on them, directing the model attention to user-specified parts. Like prompting, PASTA is applied at inference time and does not require changing any model parameters. Experiments demonstrate that PASTA can substantially enhance an LLM's ability to follow user instructions or integrate new knowledge from user inputs, leading to a significant performance improvement on a variety of tasks, e.g., an average accuracy improvement of 22% for LLAMA-7B. Our code is publicly available at `https://github.com/QingruZhang/PASTA`.

## 1 INTRODUCTION

The advent of large language models (LLMs) has marked a significant milestone in natural language processing (NLP) and artificial intelligence (AI), showcasing exceptional performance across a wide range of tasks (Vaswani et al., 2017; Brown et al., 2020a; OpenAI, 2023). Efforts to further refine these models have been relentless, aiming to enable them to process and respond to natural and programming languages with human-like expertise (Stiennon et al., 2020; Yao et al., 2023).

Despite their remarkable achievements, LLMs often encounter challenges in understanding their contextual inputs during interactions with users (Shen et al., 2023; Lu et al., 2021). This difficulty becomes particular evident when they are presented prompts[1] containing extensive background contexts or complex user instructions. Lengthy contexts can overwhelm LLMs, as their attention modules, learned from data, are unable to fully capture crucial details (Liu et al., 2023). Complex instructions can further inhibit the model from focusing on the user's intentions, resulting in undesired outputs (Wei et al., 2022). Additionally, for time-sensitive data, such as news articles, there can exist factual knowledge within contexts, which contradicts with model prior beliefs induced from outdated pre-training. As a result, a model may generate outputs conditioned on its pre-existing belief instead of attending to new facts within the contexts (Meng et al., 2022a;b; Mitchell et al., 2022; Hernandez et al., 2023). All of these challenges contribute to LLMs struggling to comprehend user intentions.

Compared to LLMs, human readers rarely struggle to understand the emphases of articles and intentions of writers. Writers often leverage a variety of text styles, such as **bold** and *italics*, to emphasize specific contents. This mechanism enables writers to direct and maintain the attention of

---

[∗]Work is done during Qingru Zhang's internship at Microsoft Research.
[1]We use *prompts* to refer to all LLM text inputs, including user instructions, and the other background information (which we refer to as *context*).

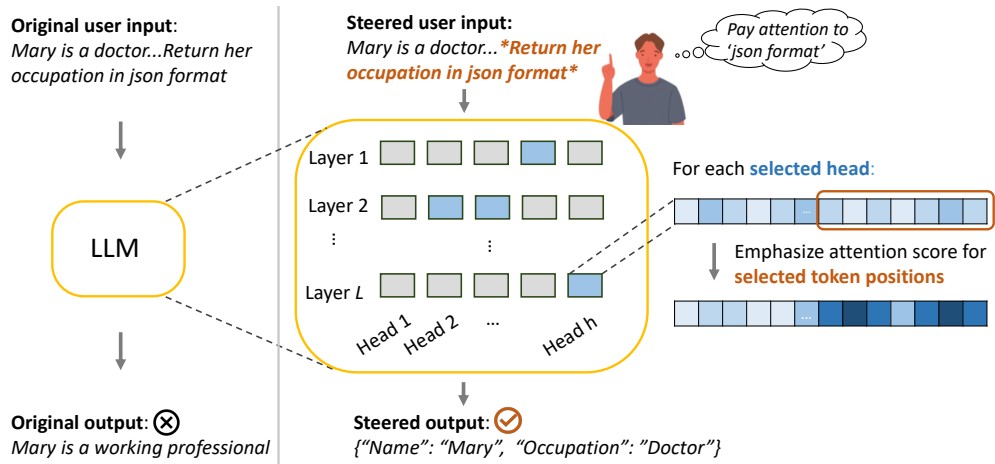

Figure 1: PASTA uses a user-specified part of the input to steer the model generation aligning with user intentions. PASTA modifies the attention scores generated during inference, by emphasizing the scores generated at token positions corresponding to the user-specified part of the context.

human readers, ensuring that the intended information is accurately captured. In interactions between users and LLMs, it is users also need to highlight specific information for the model. Consequently, model generation can be effectively biased in accordance with user guidance, thus addressing the challenges mentioned earlier. This feature is particularly essential when designing user-AI interfaces, and can be frequently applied in extensive conversations between users and models. Existing methods, however, do not support such a mechanism. LLMs are inherently limited to processing plain texts, devoid of any stylistic cues or emphasis markers (Brown et al., 2020b; Liu et al., 2021; Wei et al., 2022). Even when emphasis markers are added to prompts, state-of-the-art LLMs often struggle to discern weak signals from a couple of marker tokens (See evidence in Section 5.1).

Motivated by the need to convey user emphasis, we introduce *PASTA* (Post-hoc Attention STeering Approach), a post-hoc method[2] that enables users to highlight specific information, e.g., an instruction as in Figure 1, and steer models to interpret emphasized texts like human readers. Specifically, PASTA selects a small subset of attention heads and applies precise attention reweighting on them. As illustrated in Figure 1, PASTA upweights the attention scores of the user-specified tokens while downweighting the other tokens at specific attention heads. Our method is inspired by the observation that attention modules exhibit various token-attending patterns across different heads (Michel et al., 2019; Voita et al., 2019; Clark et al., 2019). These attention patterns can be interpreted as encoding diverse semantic or syntactic information, and altering them can substantially influence model behaviors (Shi et al., 2023a; Hu et al., 2021b). Through steering attention modules, PASTA directs the model to pay close attention to the user-specified parts and hence generate the desired output aligning with the highlighted contents. Notably, PASTA is applied after training and does not require changing any model parameters; PASTA only requires access to the attention scores of specific heads of an LLM.

Since attention heads can serve different functions (Tenney et al., 2019; Deb et al., 2023), we introduce an efficient model profiling algorithm to identify which heads are effective for steering. Specifically, we subsample small training sets from multiple tasks and evaluate the performance of attention steering for each individual head across these tasks. PASTA selects the attention heads that, when steered, generally improve the multi-task performance. We empirically observe that steering these heads not only benefits the existing tasks but also enhances the performance on unseen tasks. Notably, the model profiling is performed only once for an LLM. The selected attention heads can be regarded as a model-level profile, effective for steering the LLM on unseen tasks.

We conduct experiments on diverse tasks to demonstrate the effectiveness of PASTA. Specifically, we evaluate PASTA using GPT-J-6B (Wang & Komatsuzaki, 2021) and LLAMA-7B (Touvron et al., 2023) on tasks that span complex instructions, lengthy contexts, and knowledge conflicts within contexts. The results demonstrate that PASTA consistently provides a significant performance improvement over baseline prompting strategies. For example, PASTA achieve an average accuracy improvement of 22% over few-shot prompting for LLAMA-7B across 4 challenging tasks.

---

[2]*Post-hoc* means that our method does not update the model weights.

## 2  BACKGROUND

**Problem description**  In standard LLM prompting, we are given a pre-trained LLM and a text prompt $x$. In our setting, we additionally require (i) access to attention scores produced by attention modules in the LLM[3] and (ii) we are provided a user-specified subset of the prompt $x_g \subset x$ to be emphasized.

As in the example in Figure 1, $x$ can be a string that ends in an instruction, such as *Mary is a doctor but used to be a nurse...Return her occupation in json format*. If a user emphasizes the instruction, $x_g$ can simply be the final instruction *Return her occupation in json format*. In evaluation datasets, we assume that the user-specified part of each example is already provided by enclosing at its both ends in some emphasis markers, like '*' marker in Markdown. Generating these well-structured data often incurs little overhead. For example, in the dataset tailored for evaluting model ability to follow user instruction, we can simply mark the final instruction for every example, which are fixed and shared across examples. When it comes to user-LLM interface, users can specify $x_g$ by enclosing it with the same emphasis markers. $x_g$ can be specified flexibly. Namely, it need not be a continuous span, and can be used to emphasize diverse information.

**Multi-Head Attention.**  A typical transformer model consists of $L$ stacked layers, where each layer contains two submodules: a multi-head attention (MHA) and a fully connected feed-forward network (FFN). Given the input $X \in \mathbb{R}^{n \times d}$, MHA of the layer $l$ performs the attention function in parallel $H$ heads: $\text{MHA}^{(l)}(X) = \text{Concat}(H^{(l,1)}, ..., H^{(l,H)})W_o$ where

$$H^{(l,h)} = A^{(l,h)}V = \text{Softmax}\left(QK^\top/\sqrt{d_h}\right)V \tag{1}$$

where $Q = XW_{q_h}, K = XW_{k_h}, V = XW_{v_h}$ and $W_{q_h}, W_{k_h}, W_{v_h} \in \mathbb{R}^{d \times d_h}$ are learnable projection matrices of head $h$. $d_h$ is typically set to $d/H$. Specifically, denote the attention scores at the head $h$ of the $l$-th layer as $A^{(l,h)}$.

## 3  METHOD

PASTA (Algorithm 1) consists of two components: (i) post-hoc attention steering, which emphasizes the user-specified parts of the input during inference, see Section 3.1 and (ii) multi-task model profiling, which selects the effective attention heads for steering, see Section 3.2.

---

**Algorithm 1** PASTA: Post-hoc Attention Steering Approach

---

**Multi-task model profiling** (Section 3.2)
 1: **Input:** small training sets $\{\mathcal{D}^{(i)}\}_{i=1}^m$, the hyperparameters $\alpha$, $k$;
 2: **for** $1 \leq i \leq m$ **do**
 3:   **for** $1 \leq l \leq L, 1 \leq h \leq H$ **do**
 4:     Evaluate the model performance on $\mathcal{D}^{(i)}$ when steering the head $(l, h)$ by (2);
 5:     Return the evaluation result of steering $(l, h)$ on $\mathcal{D}^{(i)}$;
 6:   **end for**
 7:   Collect the steering results of all heads and return the task profiling $R^{(i)}$;
 8: **end for**
 9: **Output:**  The attention head set $\mathcal{H} = \cap_{i=1}^m R_{1:k}^{(i)}$.

---

**Inference-time steering** (Section 3.1)
 1: **Input:** text inputs $x$, user-underlined segments $\mathcal{G}$, coefficient $\alpha$;
 2: **Output:** the model generations while steering every head $(l, h)$ in $\mathcal{H}$ by (2).

---

### 3.1  POST-HOC ATTENTION STEERING

PASTA emphasizes the user-specified input subset by downweighting the attention scores of tokens that are not specified by the user. Specifically, given the index set of highlighted input spans as $\mathcal{G}$, PASTA emphasizes these user-specified tokens by an attention projection $\mathcal{T}$:

$$H^{(l,h)} = \mathcal{T}(A^{(l,h)})V, \text{ where } [\mathcal{T}(A)]_{ij} = \begin{cases} \alpha A_{ij}/C_i & \text{if } j \in \mathcal{G}^- \\ A_{ij}/C_i & \text{otherwise.} \end{cases} \tag{2}$$

---

[3]We do not need access model weights nor intermediate outputs from the other modules like FFNs.

where $0 \le \alpha < 1$ is a scaling coefficient and $\mathcal{G}^{-} = [n] - \mathcal{G}$ is the index set of tokens that are not in $\mathcal{G}$. The term $C_i = \sum_{j \in \mathcal{G}} \boldsymbol{A}_{ij} + \sum_{j \in \mathcal{G}^-} \alpha \boldsymbol{A}_{ij}$ normalizes the scores so that they sum to one. The attention steering (2) is conducted during the inference time and does not require any training.

(2) steers the model attention by scaling down the scores of tokens that are not highlighted by the user. When the coefficient $\alpha$ is set very small, user-specified segments are highlighted given their increased attention scores after renormalization. Consequently, we can direct the model to concentrate more on the user-specified tokens, biasing the generation to align with the specified contents.

PASTA scales down the attention scores of non-specified tokens by $\alpha$. As renormalization is followed, it is equivalent to scaling up the attention scores of user-specified tokens by $1/\alpha$. The reason of selecting (2) is that it can be more numerically stable compared to scaling up scores. Alternatively, one can also scale the attention scores by adding a positive constant to the underlined tokens $\mathcal{G}$. The reason of we select multiplication in (2) instead of addition is that it preserves the difference on attention magnitude among the highlighted tokens. As such, the steering operation only adjusts overall attention scales of two groups of tokens. In contrast, addition by a large constant to the highlighted tokens results in their attention scores almost uniformly distributed, leading to unnecessary information loss and performance degeneration.

### 3.2 Multi-Task Model Profiling

Empirically, we find that applying attention steering in (2) to all attention heads performs worse than applying it only to specific heads (see Section 5.3). It is important to specify the correct attention heads, given that different heads serve distinctive roles in encoding semantic/syntactic information. To this end, we propose a multi-task model profiling algorithm to identify the effective attention heads for steering. Specifically, given $m$ tasks involving user emphases, we subsample a small training set $\mathcal{D}^{(i)}$ (e.g., $|\mathcal{D}^{(i)}| = 1000$) from each task $i$. Then, we evaluate the performance of steering every individual attention head $(l, h)$ ($1 \le l \le L, 1 \le h \le H$) on each small subset $\mathcal{D}^{(i)}$ ($1 \le i \le m$). For every task $i$, we rank all of heads according to their steering performance on $\mathcal{D}^{(i)}$ and regard the ranking $R^{(i)} = [(l_1, h_1), (l_2, h_2), \dots]$ as the profiling of task $i$. We then set the attention head set $\mathcal{H}$ for steering as the intersection of top-$k$ performing heads, $\mathcal{H} = \cap_{i=1}^{m} R_{1:k}^{(i)}$ (see Section 5.3 for alternative choices). Intuitively, we expect performance to improve as the number of tasks $m$ increases.

Like attention steering, model profiling requires only access to attention scores, in addition to its inputs and outputs (model weights and gradients are not required). Importantly, this process needs to be performed only once for a LLM, similar to finetuning. However, unlike finetuning, model steering does not modify model weights and, more importantly, generalizes to new tasks. The resulting head set $\mathcal{H}$ can be regarded as a model-level profile. Once it is determined, we can apply the attention steering on $\mathcal{H}$ to both existing tasks and unseen tasks to enhance model contextual understanding and benefit downstream performance.

## 4 Experimental Setup

**Evaluation tasks and metrics.** We implement PASTA for two pre-trained models: GPT-J (6 billion parameters, (Wang & Komatsuzaki, 2021)) and LLaMA-7B (7 billion parameters, (Touvron et al., 2023)). We evaluate the effectiveness of PASTA at (i) handling complex user instructions, (ii) interpreting lengthy contexts, and (iii) resolving in-context knowledge conflicts. For (i), we introduce two new tasks: *JSON formatting* and *Pronouns changing*. For (ii) and (iii), we study *Bias in Bios* (De-Arteaga et al., 2019) and *CounterFact* (Meng et al., 2022a). For each task, we provide a description, describing which part of the input we emphasize, and what metrics we use for evaluation (see Appendix A for full dataset details).

● *JSON Formatting* is a new task that evaluates an LLM's ability to produce outputs in a user-desired format (JSON). This is an important usecase for LLMs when their output is being used in a downstream process. This task utilizes the biographical data from BiasBios (described below) but appends a different instruction to the end of contexts: *answer the occupation of {person} and generate the answer as JSON format*. The instruction prompts models to generate outputs in JSON format.

↪ We emphasize the final instruction

Metrics: (a) **Format accuracy** (F. Acc.) measures the accuracy at generating valid JSON. (b) **Prediction accuracy** (P. Acc.) measures the accuracy at generating the correct target in JSON values after loading the JSON-formatted generations.

● *Pronouns changing* is a new task that evaluates an LLM's ability to follow a difficult user instruction. It again uses the biographical contexts from BiasBios but instead instructs models to: *substitute 'she' and 'he' with 'they' and generate the occupation of {person} after changing pronouns*.

↪ We emphasize the final instruction.

Metrics: (a) **Accuracy** evaluates the ratio that 'she/he' are successfully changed to 'they' in model generations. (b) **All-changed accuracy** (A. Acc.) is the ratio that models replace all corresponding pronouns, i.e., changing she/he/her/him/hers/his to they/them/their/theirs.

● *CounterFact* measures an LLM's ability to generate text consistent with a new fact. Each example consists of (*subject*, *relation*, *old target*, *new target*), e.g., (*Kevin Garnett*, *is a professional*, *basketball player*, *baseball player*). We present the model both old and new facts following the prompt: *Previously, {old fact}, but currently, {new fact}. {question}*. This change in facts over time often confuses LLMs, resulting in random guesses on two of them when answering the {*question*}.

↪ We emphasize the input span containing the *new fact*.

Metrics: we evaluate metrics following (Meng et al., 2022a): (a) **Efficacy score** (ES) is the portion of cases for which the model has $P_{\mathrm{LLM}}(\text{new target}) > P_{\mathrm{LLM}}(\text{old target})$; (b) **Paraphrase score** (PS) is the same as ES but changes the {*question*} with a set of rephrased questions to assess the generalization

● *BiasBios* consists of professional biographies of non-famous people, originally introduced to investigate gender bias in occupations. Each example includes biographical context and a label of target occupation. The first sentence mentions the person's occupation, and subsequent sentences describe the individual's career history but may not be directly related to the prediction, potentially distracting the model attention. At the end of the context, we append the question: {*person*} *has the occupation of __*.

↪ We emphasize the first sentence, as it carries the most information about the occupation.

Metrics: following (Hernandez et al., 2023), we compute **Accuracy** by checking whether the probability assigned to the target occupation is the highest among the 28 candidate occupations.

For *Pronouns changing*, *CounterFact*, and *BiasBios*, we additionally measure **Fluency** as the average bi-gram and tri-gram entropy of generations, designed to be low for degenerated or repetitive texts (Meng et al., 2022a). We filter out any results receiving a fluency below 3.0 (see full results including fluency in Appendix B.1).

**Baselines.** We compare PASTA to the following baselines:

● *Zero-shot prompting* is the most common approach to interact with LLMs, in which a user feeds models a prompt containing background context and a user instruction or question.

● *Marked prompting* alters the prompts used in zero-shot prompting by surrounding user-specified input spans with emphasis markers, e.g. asterisks, as is done in markdown files for emphasis, or quotes, as is done in natural languages.

● *Few-shot prompting* includes demonstrations (example inputs and target outputs) at the beginning of the prompt fed to the LLM. Few-shot prompting often improves performance in new tasks, but increases the computational cost of inference due to the increased prompt length, particularly when demonstrations are lengthy (Dong et al., 2023); here we use 3 demonstrations in context.

**PASTA settings** We study PASTA in 2 settings: *multi-task* and *task-agnostic*. In the multi-task setting, the evaluation task $j$ is included for profiling, whereas in the task-agnostic setting, the evaluation task is excluded (instead, we profile on the 3 datasets besides $j$). The multi-task setting improves performance but requires labeled training samples for the task which is evaluated, which can be difficult to obtain in practice.

Empirically, we find that PASTA is not sensitive to the scaling coefficient $\alpha$ (see Section 5.3) and fix it to 0.01 in our experiments. We select 1000 training samples from each of the 4 tasks above for model profiling. After model profiling, we select $k$ from {300, 400, 500} for LLAMA-7B

Table 1: Main results of LLAMA-7B to demonstrate that PASTA can improve the model ability to (i) follow user instruction (*JSON Format* and *Prons. Changing*); (ii) interpret contextual information (*BiasBios*); (iii) resolving knowledge conflicts (*CounterFact*). For all scores, higher is better. The best results are in **bold**.

|  | Method | JSON Format | Prons. Changing | BiasBios | CounterFact | All |
|---|---|---|---|---|---|---|
|  |  | F. Acc / P. Acc | Acc / A.Acc | Acc | ES / PS | Ave. |
| **Prompting** | Zero-shot | 60.00 / 54.94 | 71.84 / 66.28 | 87.36 | 58.50 / 52.03 | 67.29 |
|  | ∗-marked | 18.55 / 12.71 | 39.14 / 35.17 | 90.62 | 57.74 / 50.52 | 49.38 |
|  | ""-marked | 4.56 / 4.20 | 20.55 / 18.19 | 89.82 | 58.14 / 51.70 | 42.15 |
|  | Few-shot | 84.85 / 73.58 | 59.06 / 55.27 | 88.79 | 87.45 / 49.82 | 73.45 |
| **PASTA** | Task-agnostic | 88.16 / 49.08 | 83.65 / 81.31 | 93.54 | 98.82 / 99.03 | 85.89 |
|  | Multi-task | **96.64 / 85.09** | **96.42 / 95.84** | **95.28** | **99.60 / 99.57** | **95.46** |

Table 2: Main results of GPT-J to demonstrate that PASTA can improve the model ability to (i) follow user instruction (*JSON Format* and *Prons. Changing*); (ii) interpret contextual information (*BiasBios*); (iii) resolving knowledge conflicts (*CounterFact*). For all scores, higher is better. The best results are in **bold**.

|  | Method | JSON Format | Prons. Changing | BiasBios | CounterFact | All |
|---|---|---|---|---|---|---|
|  |  | F. Acc / P. Acc | Acc / A.Acc | Acc | ES / PS | Ave. |
| **Prompting** | Zero-shot | 28.83 / 25.09 | 39.88 / 36.19 | 72.76 | 42.14 / 42.02 | 44.96 |
|  | ∗-marked | 4.44 / 4.10 | 41.25 / 37.57 | 74.14 | 44.50 / 45.09 | 40.63 |
|  | ""-marked | 8.81 / 5.62 | 6.12 / 5.72 | 78.64 | 45.54 / 41.84 | 33.87 |
|  | Few-shot | 84.15 / **72.65** | 35.77 / 32.08 | 72.98 | 68.34 / 38.23 | 59.65 |
| **PASTA** | Task-agnostic | 46.68 / 34.71 | 91.62 / 88.60 | 80.84 | **99.54 / 99.57** | 77.80 |
|  | Multi-task | **91.50** / 18.63 | **92.96 / 91.34** | **94.96** | 98.62 / 98.79 | **85.22** |

to have the number of steered heads $|\mathcal{H}|$ as $\{25, 53, 86\}$. We find that PASTA achieves the best performance on LLAMA-7B when $50 \leq |\mathcal{H}| \leq 100$, i.e., $k = 400$ or $k = 500$. For GPT-J, we select $k$ from $\{250, 275, 300, 350\}$ to have $|\mathcal{H}|$ as $\{52, 72, 111, 153\}$. For every task, we split data into train/validation/test sets following (Hernandez et al., 2023) (See Appendix A) and select $|\mathcal{H}|$ by cross validation. For all tasks, model outputs are generated with greedy search.

## 5 RESULTS

### 5.1 MAIN RESULT: PASTA IMPROVES MODEL GENERATION

Tables 1 and 2 present the main results for PASTA applied to LLAMA-7B and GPT-J respectively. Few-shot prompting is the strongest baseline, and task-agnostic PASTA outperforms it on the main metric for each task for all settings except JSON Formatting with GPT-J. Multi-task PASTA outperforms all baselines across all settings.

**PASTA can improve LLM instruction following.** The results from JSON Formatting and Pronouns Changing tasks indicate that, by highlighting the user instruction at the end of inputs, PASTA effectively steers models to focus on user intentions, thereby biasing their generation to fulfill specific requirements or formats. For example, while GPT-J only achieves 39.9% of its zero-shot generations complying the user requirement on the Pronouns Changing task, PASTA yields a remarkable 53% accuracy improvement by emphasizing the instruction. Moreover, PASTA achieves an impressive 96.64% format accuracy and 85.09% prediction accuracy when applied to LLAMA-7B on the JSON Formatting task. This performance exceeds that of few-shot prompting by 11%, even though few-shot prompting explicitly provides the model with correct JSON examples through additional demonstrations. Table 3 presents a few examples generated by LLAMA-7B when applying PASTA.

**PASTA can help models capture crucial contextual information.** In the case of BiasBios and CounterFact tasks, we apply PASTA to emphasize specific context spans for LLMs. Consequently, the models are guided to pay close attention to the specific contextual information or new facts within

Table 3: Generation examples of LLAMA-7B on JSON Formatting and Pronouns Changing tasks.

| Task | Prompt | Zero-shot Generation | PASTA Generation |
|---|---|---|---|
| JSON Format | Winnie is an American photographer living in New York. Specialized in fashion photography and portrait, she applies her talent on both humans and animals ... {instruction} | ✗ Winnie is a fashion and portrait photographer living in New York, specialized in fashion photography and portrait, who applies her talent on both humans and animals. | ✓ {"name": "Winnie", "occupation": "photographer" } |
| Prons. Change | Mary is an Associate Professor in the Department of Curriculum Instruction at St. John University; she holds a doctorate in Reading/Writing/Literacy from the University of Pennsylvania. She recently published an article in "Urban Education" with ... {instruction} | ✗ Mary is a professor in the Department of Curriculum Instruction at St. John University; **she** holds a doctorate in Reading/Writing/Literacy from the University of Pennsylvania. **She** recently published an article in "Urban Education" with ... | ✓ Mary is an associate professor in the department of curriculum and instruction at St. John's University; **they** hold a doctorate in reading/writing/literacy from the University of Pennsylvania. **They** recently published an article in ... |

contexts. The results from these two tasks illustrate that PASTA can direct the models to interpret the crucial information or resolve the knowledge conflicts within contexts, yielding significant improvement in prediction performance of both tasks. For example, PASTA achieves a prediction accuracy of 94.96% for GPT-J on the BiasBios task, which is 16.32% higher than the best baseline.

Tables 1 and 2 also suggest that marked prompting, a baseline that highlights specific texts akin to human writers, struggles to effectively convey emphasis to LLMs. One possible reason is that these emphasis markers rarely appear in the massive pre-training data. In contrast, few-shot prompting sometimes leads to improvements in model performance. However, a drawback of few-shot prompting is its instability, i.e. its performance exhibits high variance across different samples in the demonstration (See Appendix B).

## 5.2 PASTA CAN MITIGATE THE SENSITIVITY OF PROMPTS

Table 4: Results about sensitivity of model performance to prompt rephrasing on the JSON Formatting task. Given rephrased instructions in prompt template, PASTA can imporve zero-shot performance for all prompts.

| Instruction | Method | LLAMA-7B | | GPT-J | | Average |
|---|---|---|---|---|---|---|
| | | JSON Format F. Acc / P. Acc | Prons. Changing Acc / A. Acc | JSON Format F. Acc / P. Acc | Prons. Changing Acc / A. Acc | |
| Original | Zero-shot | 60.0 / 54.9 | 71.8 / 66.3 | 28.8 / 25.1 | 39.9 / 36.2 | 47.9 |
| | **PASTA** | 96.6 / 85.1 | 96.4 / 95.8 | 91.5 / 18.6 | 93.0 / 91.3 | 83.5 |
| Shortened | Zero-shot | 36.0 / 32.4 | 49.2 / 42.6 | 25.4 / 17.1 | 56.5 / 54.8 | 39.3 |
| | **PASTA** | 87.4 / 65.9 | 89.0 / 86.9 | 54.1 / 37.0 | 94.0 / 93.7 | 76.0 |
| Rephrased | Zero-shot | 57.9 / 54.2 | 82.3 / 79.6 | 63.3 / 50.3 | 76.0 / 72.8 | 67.1 |
| | **PASTA** | 97.1 / 87.1 | 89.6 / 89.0 | 77.5 / 68.1 | 94.8 / 92.3 | 86.9 |

It is well-known that the the performance of LLMs can be sensitive to minor changes in prompts, such as rephrasing and reformatting, even when these prompts convey the same meaning (Reynolds & McDonell, 2021; Liu et al., 2021). We find that PASTA can alleviate the sensitivity of model performance to varying prompts. Specifically, Table 4 evaluates the performance of LLAMA-7B and GPT-J on JSON Formatting and Pronouns Changing task given different instructions in the prompt template, all of which convey the same meaning (see precise prompts in Appendix A.1). The results show that zero-shot performance is sensitive to different prompts and can significantly deteriorate with poorly crafted templates. In contrast, PASTA consistently improves model performance over zero-shot prompting for all prompts, effectively mitigating sensitivity to variations in the prompts.

## 5.3 ANALYSIS AND ABLATIONS

In this section, we investigate different hyperparameter choices and modeling decisions that affect the performance of PASTA.

**Model profiling** Figure 2 presents the results on the importance of model profiling introduced in Section 3.2. We compare PASTA when steering the selected heads versus other reasonable choices: steering (i) all heads, (ii) entire layers, or (iii) individual heads on the JSON Formatting task (See

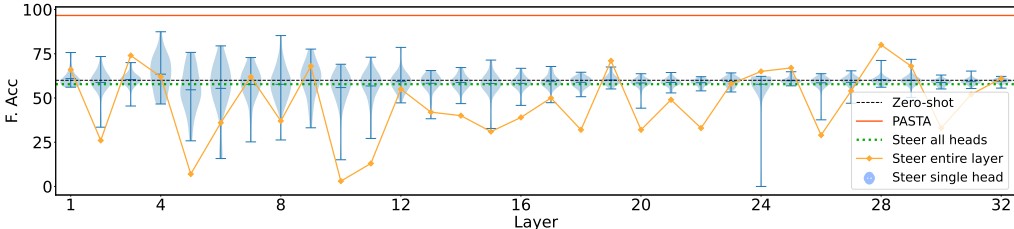

Figure 2: The performance of LLAMA-7B on the JSON Formatting task when we steer (i) all heads (green); (ii) an entire layer (yellow); and (iii) an individual head within a layer (blue violin plot). The performance varies dramatically across layers and across heads of a layer.

Appendix B.3 for comparisons on the remaining tasks). Selecting heads via model profiling in PASTA (red line) significantly outperforms other approaches. Steering all heads (dashed green line) degrades performance compared to the baseline zero-shot performance (dashed black line). This is likely because steering all heads over-amplifies the user-specified information at the expense of other essential information required for effective generation and prediction. Interestingly, we find that the performance varies significantly when steering different layers (yellow) or heads (blue violin plot). As mentioned in Section 1, attention heads play distinct roles in encoding diverse semantic and syntactic information (Tenney et al., 2019). When steering heads, which are appropriately involved in encoding of user-specified information, the model can be guided to capture and reinforce these specific signals. Conversely, modifying the attention of unrelated heads not only fails to emphasize the desired information but also interferes with their original functions, resulting in performance deterioration. Therefore, it is important to identify the effective heads through model profiling prior to applying the steering.

**Varying strategies for selecting heads during profiling.** As described in Sec. 5.3, our model profiling selects the *Intersection* of the top-$k$ performing heads to steer across multiple tasks. Alternatively, when evaluating on task $j$, we can select heads for steering with different strategies: (i) *Task-specific* – steer the top-$k_2$ performing heads of only the task $j$, i.e., $R_{1:k_2}^{(j)}$; or (ii) *Union* – the union of these heads across multiple tasks, i.e., $\cup_{i=1}^{m} R_{1:k_2}^{(i)}$. Table 5 compares their performance. Using task-specific heads rather than intersection-selected heads sometimes yields improved performance, but requires selecting a different set of heads for each new task.

Table 5: Varying head selection strategies between top *task-specific* heads, *union* across multiple tasks, and intersection (the default used in PASTA).

| | **PASTA** | **JSON Format** F. Acc / P. Acc | **Prons. Changing** Acc / A.Acc | **BiasBios** Acc | **CounterFact** ES / PS | **All** Avg. |
|---|---|---|---|---|---|---|
| **LLAMA** | Task-specific | 95.56 / 86.83 | 98.52 / 98.02 | 97.62 | 99.18 / 99.24 | 96.57 |
| | Union | 88.42 / 74.49 | 92.12 / 91.44 | 96.36 | 99.24 / 99.35 | 92.22 |
| | Intersection | 96.64 / 85.09 | 96.42 / 95.84 | 95.28 | 99.60 / 99.57 | 95.46 |
| **GPT-J** | Task-specific | 85.71 / 79.39 | 94.74 / 92.54 | 97.64 | 99.26 / 99.34 | 93.29 |
| | Union | 72.61 / 64.89 | 89.68 / 87.76 | 95.56 | 99.82 / 99.83 | 88.21 |
| | Intersection | 91.50 / 18.63 | 92.96 / 91.34 | 94.96 | 98.62 / 98.79 | 85.22 |

**Varying the number of heads to be steered.** Figures 3a and 3b illustrate the performance of PASTA when steering different number of heads on two tasks. The results suggest that as more heads are included for steering, the model follows the user even more closely, achieving higher efficacy (JSON Format Acc. and Pron. Change Acc.). However, at some point, this it results in a decrease in the metrics reflecting the generation quality (JSON Pred. Acc and Fluency). Thus, there is a trade-off between emphasizing efficacy and generation quality. Overemphasizing can lead the model to focus solely on satisfying the user requirements and ignore the other parts. Therefore, we recommend applying PASTA to a moderate number of heads (typically 50 to 150), striking a balance between the efficacy and generation quality.

**Varying the scaling coefficient $\alpha$.** Figure 3c presents the performance of PASTA on two tasks when we change the scaling coefficient $\alpha$. The results indicate that PASTA is fairly robust to this

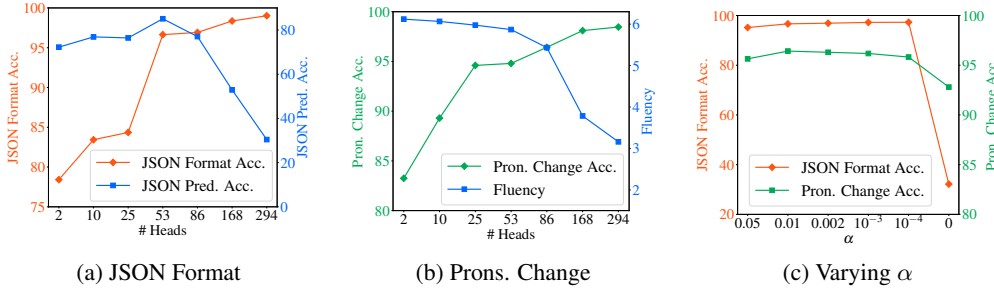

(a) JSON Format   (b) Prons. Change   (c) Varying $\alpha$

Figure 3: The performance of applying PASTA to LLAMA-7B on JSON Formating and Pronouns Changing tasks when varying the number of steered heads $|\mathcal{H}|$ (3a,3b); and changing the scaling coefficient $\alpha$ (3c).

hyperparameter; in practice, we fix it as 0.01. Notice that setting $\alpha$ to zero should be avoided, as this leads to the complete removal of other crucial contexts at the steered heads, resulting in performance degeneration.

# 6  RELATED WORK

The primary method for controlling LLMs has been through prompting, often yielding impressive improvements in performance (Brown et al., 2020b; Liu et al., 2021; Wei et al., 2022) and spurring a line of work aiming to make prompting easier, e.g. (Strobelt et al., 2022; Bach et al., 2022; Shin et al., 2020; Deng et al., 2022; Singh et al., 2023b). However, LLMs remain extremely sensitive to nuances in prompts (Webson & Pavlick, 2021; Lu et al., 2021); PASTA complements these approaches by making it easier for a user to specify a prompt in difficult scenarios.

Another line of work aims to make LLMs more amenable to prompting by modifying them during training. Most prominent among these approaches are instruction finetuning (Wei et al., 2021; Chung et al., 2022), Reinforcement Learning from Human Feedback (Ziegler et al., 2019; Ouyang et al., 2022), and other related methods, e.g. (Lee et al., 2023). There are also a few methods for directly specifying which parts on an input are important during training, e.g. (Ross et al., 2017; Rieger et al., 2019; Schramowski et al., 2020; Krishna et al., 2023). PASTA can be used in addition to these approaches to improve some aspects of model steerability (e.g. instruction following).

PASTA is related to variety of methods for adapting to new tasks, including LoRA (Hu et al., 2021a), AdaLoRA (Zhang et al., 2023), QLoRA (Dettmers et al., 2023), and TOAST (Shi et al., 2023b). PASTA is also related to a variety of research on model editing, e.g. ROME (Meng et al., 2022a), MEMIT (Meng et al., 2022b), MEND (Mitchell et al., 2022), and REMEDI (Hernandez et al., 2023). Unlike these works, PASTA preserves an LLMs ability to transfer to new tasks using prompts and human-selected info, rather than using new labeled examples.

Finally, PASTA is also motivated by works which have aimed to mechanistically understand attention scores (Zou et al., 2023), e.g. by studying them through feature importance (Jain & Wallace, 2019; Wiegreffe & Pinter, 2019; Deb et al., 2023), probing (Conneau et al., 2018; Liu & Avci, 2019), visualization (Karpathy et al., 2015; Olah et al., 2017), localizing knowledge (Meng et al., 2022a; Dai et al., 2021), categorizing directions in representation space (Kim et al., 2017; Schwettmann et al., 2021), or natural-language explanations (Bills et al., 2023; Singh et al., 2023a).

# 7  CONCLUSION

In this study, we propose PASTA, a novel approach aimed at enabling LLMs to move beyond the limitations of plain text and effectively perceive user guidance embodied as highlighted parts of prompts. By making precise adjustments to attention scores in selected heads, PASTA directs the model's focus to the relevant context, mirroring the way humans benefit from textual cues. Unlike traditional fine-tuning methods, PASTA is applied at inference time and requires neither parameter updates nor gradient computation; PASTA requires only selecting which attention heads to apply the re-weighting to, a one-time profiling operation for a LLM. Experimental results show that PASTA can significantly improve model performance on a variety of tasks. In the future, we plan to integrate PASTA with various other methods, such as few-shot in-context learning, aiming to highlight effective examples to enhance its stability.

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

# APPENDIX

## A    EXPERIMENTAL DETAILS

We implement all algorithms using `PyTorch` (Paszke et al., 2019) and `Huggingface` (Wolf et al., 2019) and run experiments on NVIDIA V100 GPUs and NVIDIA A6000 GPUs.

Table 6 provides detailed statistics of datasets in our experiments.

Table 6: Statistics of datasets.

| Task | Train | Valid | Test |
|---|---|---|---|
| CounterFact | 1000 | 1000 | 5000 |
| BiasBios | 1000 | 1000 | 5000 |
| JSON Formatting | 1000 | 1000 | 5000 |
| Pronouns Changing | 1000 | 1000 | 5000 |

### A.1    DETAILED PROMPT TEMPLATES OF EACH TASK

For each task, the prompt templates in our results are as follows:

- **JSON Formatting**:
    - (Original) {*context*}. *Answer the occupation of* {*person*} *and generate the answer as json format. Here is an example:* {*"name": , "occupation": ,*}. *Now generate the answer.*
    - (Shortened one in Section 5.2) {*context*}. *Answer the occupation of* {*person*} *and generate the answer as json format.*
    - (Rephrased one in Section 5.2) *Answer the occupation of* {*person*} *and generate the answer as json format. Here is an example:* {*"name": , "occupation": ,*}. {*context*}. *Now generate the answer.*

- **Pronouns Changing:**
    - (Original): {*context*}. *For the aforementioned text, substitute 'she' and 'he' with 'they' and generate the occupation of* {*person*} *after changing pronouns.*
    - (Shortened one in Section 5.2): {*context*}. *Change 'she' and 'he' with 'they' and answer the occupation of* {*person*} *after replacing the pronouns*
    - (Rephrased one in Section 5.2): {*context*}. *For the aforementioned descriptions, replace 'she' and 'he' with 'they' in the aformentioned text and generate the new text after replacing the pronouns.*

- **BiasBios:** {*context*}. {*person*} *has the occupation of.*

- **CounterFact:** *Previously,* {*old fact*}. *Currently,* {*new fact*}. {*question*}

### A.2    THE EVALUATION DETAILS OF PASTA

Table 7 presents the number of heads to be steered by PASTA for LLAMA-7B and GPT-J-6B on every task.

Table 7: The number of heads to be steered by PASTA.

| Task | LLAMA-7B | GPT-J-6B |
|---|---|---|
| JSON Formatting | 53 | 153 |
| Pronouns Changing | 86 | 72 |
| BiasBios | 86 | 111 |
| CounterFact | 86 | 52 |

# B EXTENDED RESULTS

## B.1 EXTENDED RESULTS WITH FLUENCY

In this section, we include extended results, including fluency metrics. Fluency score is the average bi-gram and tri-gram entropy of generations, designed to be low for degenerated or repetitive texts (Meng et al., 2022a). This metric can be regarded as the reference metric of generation quality. Typically, the generations of language models are reliable as long as their fluency score is not too low. Here, we filter out any results receiving a fluency score below 3.0. Table 8, 9 and 10 include all results and fluency evaluation.

Table 8: Main results of LLAMA-7B to demonstrate that PASTA can improve the model ability to (i) follow user instruction (*JSON Format* and *Prons. Changing*); (ii) interpret contextual information (*BiasBios*); (iii) resolving knowledge conflicts (*CounterFact*). For all scores, higher is better. The best results are in **bold**.

| | Method | JSON Format
F. Acc / P. Acc | Prons. Changing
Acc / A.Acc / Flue. | BiasBios
Acc / Flue. | CounterFact
ES / PS /Flue. |
|---|---|---|---|---|---|
| **Prompting** | Zero-shot | 60.00 / 54.94 | 71.84 / 66.28 / **6.10** | 87.36 / 3.98 | 58.50 / 52.03 / 4.96 |
| | ∗-marked | 18.55 / 12.71 | 39.14 / 35.17 / 6.03 | 90.62 / 3.89 | 57.74 / 50.52 / 5.12 |
| | ""-marked | 4.56 / 4.20 | 20.55 / 18.19 / 5.13 | 89.82 / 3.97 | 58.14 / 51.70 / 5.13 |
| | Few-shot | 84.85 / 73.58 | 59.06 / 55.27 / 5.95 | 88.79 / **4.19** | 87.45 / 49.82 / **5.68** |
| **PASTA** | Task-agnostic | 88.16 / 49.08 | 83.65 / 81.31 / 4.62 | 93.54 / 3.03 | 98.82 / 99.03 / 4.78 |
| | Multi-task | **96.64 / 85.09** | **96.42 / 95.84** / 5.43 | **95.28** / 4.05 | **99.60 / 99.57** / 4.89 |

Table 9: Main results of GPT-J to demonstrate that PASTA can improve the model ability to (i) follow user instruction (*JSON Format* and *Prons. Changing*); (ii) interpret contextual information (*BiasBios*); (iii) resolving knowledge conflicts (*CounterFact*). For all scores, higher is better. The best results are in **bold**.

| | Method | JSON Format
F. Acc / P. Acc | Prons. Changing
Acc / A.Acc / Flue. | BiasBios
Acc / Flue. | CounterFact
ES / PS /Flue. |
|---|---|---|---|---|---|
| **Prompting** | Zero-shot | 28.83 / 25.09 | 39.88 / 36.19 / 5.91 | 72.76 / **5.06** | 42.14 / 42.02 / 5.01 |
| | ∗-marked | 4.44 / 4.10 | 41.25 / 37.57 / 4.76 | 74.14 / 5.01 | 44.50 / 45.09 / 5.22 |
| | ""-marked | 8.81 / 5.62 | 6.12 / 5.72 / 5.43 | 78.64 / 4.96 | 45.54 / 41.84 / 5.16 |
| | Few-shot | 84.15 / **72.65** | 35.77 / 32.08 / **6.46** | 72.98 / 4.82 | 68.34 / 38.23 / **5.67** |
| **PASTA** | Task-agnostic | 46.68 / 34.71 | 91.62 / 88.60 / 3.00 | 80.84 / 4.92 | **99.54 / 99.57** / 5.11 |
| | Multi-task | **91.50** / 18.63 | **92.96 / 91.34** / 4.91 | **94.96** / 4.87 | 98.62 / 98.79 / 5.11 |

Table 10: Varying head selection strategies between top top *task-specific* heads, *union* across multiple tasks, and intersection (the default used in PASTA).

| | PASTA | JSON Format
F. Acc / P. Acc | Prons. Changing
Acc / A.Acc / Flue. | BiasBios
Acc / Flue. | CounterFact
ES / PS /Flue. |
|---|---|---|---|---|---|
| **LLAMA** | Task-specific | 95.56 / 86.83 | 98.52 / 98.02 / 5.92 | 97.62 / 4.18 | 99.18 / 99.24 / 4.93 |
| | union | 88.42 / 74.49 | 92.12 / 91.44 / 4.88 | 96.36 / 4.13 | 99.24 / 99.35 / 4.53 |
| | intersection | 96.64 / 85.09 | 96.42 / 95.84 / 5.43 | 95.28 / 4.05 | 99.60 / 99.57 / 4.89 |
| **GPT-J** | Task-specific | 85.71 / 79.39 | 94.74 / 92.54 / 5.07 | 97.64 / 5.06 | 99.26 / 99.34 / 4.94 |
| | Union | 72.61 / 64.89 | 89.68 / 87.76 / 3.92 | 95.56 / 5.02 | 99.82 / 99.83 / 5.03 |
| | Intersection | 91.50 / 18.63 | 92.96 / 91.34 / 4.91 | 94.96 / 4.87 | 98.62 / 98.79 / 5.11 |

## B.2 THE VARIANCE OF FEW-SHOT PERFORMANCE

Few-shot prompting sometimes leads to improvements in model performance. as explicitly providing the examples in additional demonstrations. However, a drawback of few-shot prompting is its insta-

bility, meaning its performance exhibits high variance across different samples in the demonstratio. In this section, we present the results to show that the performance of few-shot prompting displays high variance in terms of sampling different few-shot demonstrations.

Table 11: The few-shot performance (Acc. / A. Acc. / Fluency) on the Pronouns Changing task.

| Few-shot examples | LLAMA-7B | GPT-J-6B |
|---|---|---|
| Demonstration 1 | 84.87 / 90.09 / 4.74 | 43.82 / 40.36 / 6.43 |
| Demonstration 2 | 57.24 / 53.98 / 6.22 | 40.68 / 37.86 / 6.44 |
| Demonstration 3 | 57.08 / 53.22 / 6.02 | 33.13 / 29.21 / 6.48 |
| Demonstration 4 | 52.26 / 48.30 / 6.42 | 25.47 / 20.89 / 6.44 |
| Demonstration 5 | 43.86 / 40.78 / 6.43 | 11.90 / 8.63 / 6.51 |

## B.3 MODEL PROFILING RESULTS

In this Section, we provide more results of the performance of LLAMA-7B on all of tasks when steering: (i) all heads; (ii) entire layer; (iii) a individual head of a layer.

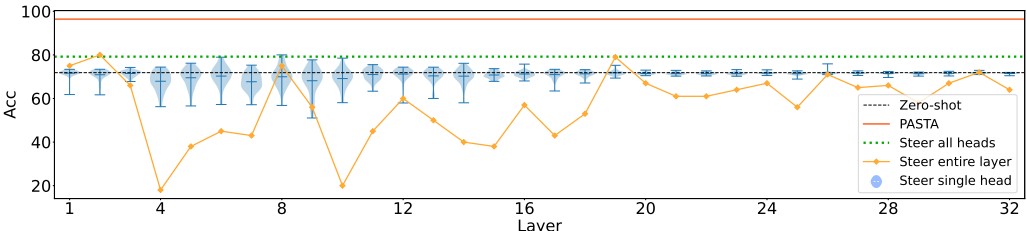

Figure 4: The performance of LLAMA-7B on Pronouns Changing task when we steer (i) all heads (green); (ii) entrie layer (yellow); and (iii) individual head with a layer (blue violin plot). The performance varies dramatically across layers and across heads of a layer.

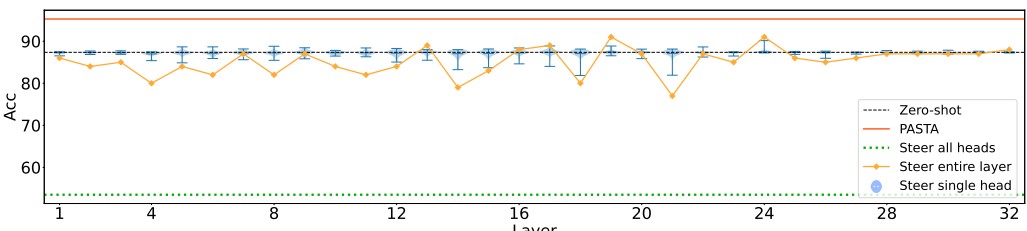

Figure 5: The performance of LLAMA-7B on BiasBios task when we steer (i) all heads (green); (ii) entrie layer (yellow); and (iii) individual head with a layer (blue violin plot). The performance varies dramatically across layers and across heads of a layer.

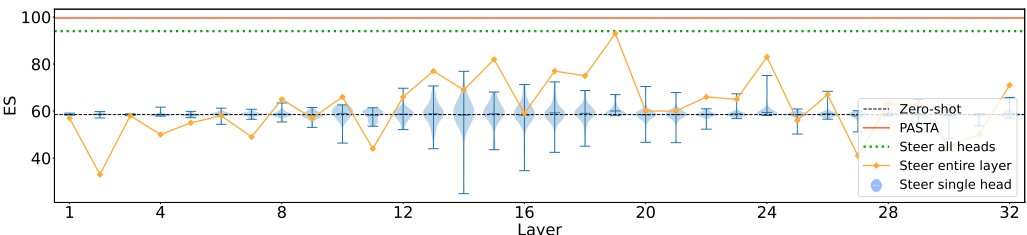

Figure 6: The performance of LLAMA-7B on CounterFact task when we steer (i) all heads (green); (ii) entrie layer (yellow); and (iii) individual head with a layer (blue violin plot). The performance varies dramatically across layers and across heads of a layer.

## C RESULTS ON MORE MODELS

### C.1 LARGER MODEL SIZE

We conduct experiments with LLAMA-13B to further evaluate the effectiveness of PASTA across all tasks. The following table presents the performance comparison for LLAMA-13B.

Table 12: Results of LLAMA-13B. For all scores, higher is better. The best results are in **bold**.

| Method | JSON Format F. Acc / P. Acc | Prons. Changing Acc / A.Acc | BiasBios Acc | CounterFact ES / PS | All Ave. |
|---|---|---|---|---|---|
| Zero-shot prompting | 45.48 / 43.16 | 65.03 / 60.90 | 85.80 | 47.86 / 44.14 | 56.05 |
| Few-shot prompting | 39.80 / 3.56 | 82.33 / 80.71 | 88.38 | 90.63 / 65.49 | 64.41 |
| **PASTA** (Multi-task) | **98.74 / 89.88** | **97.56 / 96.78** | **95.34** | **99.38 / 99.30** | **96.71** |

### C.2 STEERING INSTRUCTION-TUNED MODELS WITHOUT RE-PROFILING

We further test PASTA's applicability to Vicuna-7B-v1.3, which is instruction-tuned from LLAMA-7B. We apply PASTA using attention heads selected from LLAMA-7B profiling (including multi-task and task-specific heads). In this way, we evaluate if the heads selected from the base models are transferable to an instruction-tuned model, thereby avoiding the re-profiling. The table below presents the performance of Vicuna across all tasks.

Table 13: Results of Vicuna-7B. For all scores, higher is better. The best results are in **bold**.

| Method | JSON Format F. Acc / P. Acc | Prons. Changing Acc / A.Acc | BiasBios Acc | CounterFact ES / PS |
|---|---|---|---|---|
| LLAMA-7B Zero-shot | 60.00 / 54.94 | 71.84 / 66.28 | 87.36 | 58.50 / 52.03 |
| LLAMA-7B PASTA(multi-task) | 96.64 / 85.09 | 96.42 / 95.84 | 95.28 | 99.60 / 99.57 |
| Vicuna Zero-shot | 65.41 / 61.78 | 95.74 / 94.74 | 90.74 | 61.10 / 52.46 |
| Vicuna PASTA(multi-task) | 66.09 / 56.00 | 98.82 / 98.08 | 96.44 | 99.80 / 99.80 |
| Vicuna PASTA(task-specific) | 90.54 / 86.53 | 98.62 / 98.04 | 97.42 | 99.82 / 99.74 |

The results demonstrate that the attention heads selected for LLAMA-7B effectively steer Vicuna-7B, indicating that re-profiling is not necessary for instruction-tuned models. Notably, when steering task-specific heads selected from LLAMA profiling, PASTA significantly enhances Vicuna's performance across all tasks. This evidence shows that PASTA can complement instruction tuning without necessitating re-profiling.

### C.3 ABLATION ABOUT THE NUMBER OF EXAMPLES FOR PROFILING

the robustness of head performance ranking to sample variance allows us to further reduce the sample size for profiling (e.g., $|\mathcal{D}| = 200$). The table below presents the PASTA performance on the JSON Formatting task when re-profiling with $|\mathcal{D}| = 200$ samples. We can see PASTA still achieves superior performance when profiling with much fewer examples.

Table 14: The performance of PASTA with different sample size $|\mathcal{D}|$ of model profiling.

| Model | Sample size | JSON Format Acc | JSON Pred. Acc |
|---|---|---|---|
| LLAMA-7B Zero-shot | N/A | 60.00 | 54.94 |
| LLAMA-7B w. PASTA | 1000 | 95.56 | 86.83 |
| LLAMA-7B w. PASTA | 200 | 95.04 | 84.17 |
| Vicuna-7B Zero-shot | N/A | 65.41 | 61.78 |
| Vicuna-7B w. PASTA | 200 | 98.56 | 94.56 |

# D EXAMPLES OF PASTA GENERATIONS

Table 15: Generation examples of LLAMA-7B on JSON Formatting.

| Prompt | PASTA Generation |
|---|---|
| Winnie is an American photographer living in New York. Specialized in fashion photography and portrait, she applies her talent on both humans and animals. In this series of pictures, It's all about personification these dogs in the spotlight. The whole thing gives a great result. More pictures & infos on her portfolio. {instruction} | {"name": "Winnie", "occupation": "photographer" }\n{"name": "Winnie", "occupation": "photographer" }... |
| Jon is a software engineer on the Core Processing team at comScore, an online audience measurement and analytics company. He prefers to do most of his coding in Pig. Before working at comScore he wrote software to optimize supply chains and to allocate fixed income securities. {instruction} | {"name": "Jon", "occupation": "Software Engineer"}\n{"name": "Jon", "occupation": "Software Engineer"}... |
| Juris is an associate professor at the Faculty of Information, University of Toronto. He is a co-author of Readers Advisory Service in North American Public Libraries, 1870-2005 (McFarland, 2007); The Evolution of Library and Museum Partnerships: Historical Antecedents, Contemporary Manifestations, and Future Directions (Libraries Unlimited, 2004); and Reading and the Reference Librarian: The Importance to Library Service of Staff Reading Habits (McFarland, 2004).{instruction} | {"name": "Juris", "occupation": "Associate Professor"} \n{"name": "Juris", "occupation": "Associate Professor"}... |
| Gregory is an educational psychologist whose major concern is with how people learn, ie skills and knowledge development, especially in educational settings. His emphasis is on a strong research (statistical) foundation crossing areas such as social learning theory, information processing, and a cognitive approach to emotional factors. {instruction} | {"name": "Gregory", "occupation": "Educational Psychologist"}\n{"name": "Gregory", "occupation": "Educational Psychologist"}... |
| Asif is a Solutions Architect with Amazon Web Services. He provides technical guidance, design advice and thought leadership to some of the largest and successful AWS customers and partners on the planet. His deepest expertise spans application architecture, containers, devops, security, machine learning and SaaS business applications. Over the last 12 years, he's brought an intense customer focus to challenging and deeply technical roles in multiple industries. He has a number of patents and has successfully led product development, architecture and customer engagements. {instruction} | {"name": "Asif", "occupation": "Solutions Architect"}\n{"name": "Asif", "occupation": "Solutions Architect"}... |
| Graham is an Associate Professor in Construction Management at the University of Newcastle, Australia, where he was first appointed at the end of 1999. He is currently Deputy Head of the School of Architecture and Built Environment, and was previously Head of Discipline, Building. During this time he has undertaken various administrative roles at School, Faculty, and University level, and is currently a member of Academic Senate. {instruction} | {"name": "Graham", "occupation": "Associate Professor in Construction Management"}\n{"name": "Graham", "occupation": "Associate Professor in Construction Management"}... |
| Katherine is an Assistant Professor in the Department of Pathology. She received her MD from the University of Colorado School of Medicine where she also completed her AP/CP residency training in pathology and a fellowship in Cytogenetics. She completed her fellowship in Molecular Genetic Pathology in the Department of Pathology in 2009. She is a Medical Director in Cytogenetics, Molecular Cytogenetics, and Molecular Oncology at ARUP Laboratories. {instruction} | {"name": "Katherine", "occupation": "Assistant Professor in the Department of Pathology", "answer": "Katherine is an Assistant Professor in the Department of Pathology. She received her MD from the University of Colorado School of Medicine where she also completed her AP/CP residency training in pathology and a fellowship in Cytogenetics.", |
| Peter is an immigration attorney with at least three links to the Mexican government. In addition to heading the Center for Human Rights and Constitutional Law, he also operates vocesunidas.org, which until recently was called the "Mexico Project". It's since undergone an interesting change. {instruction} | {"name": "Peter", "occupation": "Immigration Attorney"}\n The answer is a json object. {"name": "Peter", "occupation": "Immigration Attorney"}\n{"name": "Peter", "occupation": "Immigration Attorney"}... |
| Delvin is a freelance journalist. He was also the Turtle Mountain Band of Chippewa's Oil and Gas Specialist. He is a frequent contributor to the tribe's newspaper, The Turtle Mountain Times, as well as indianz.com and Indian Country Today Media Network, both national news outlets for Indian Country. {instruction} | {"name": "Delvin", "occupation": "Freelance Journalist"}\n {"name": "Delvin", "occupation": "Freelance Journalist", "age": 25}\n {"name": "Delvin", "occupation": "Freelance Journalist", "age": 25, "gender": "male"}\n {"name": "Delvin", "occupation": "Freelance Journalist", "age": 25, "gender": "male", "hair": "black", "eyes": "brown"}... |
| Robert is a rock photographer based in Los Angeles. He started his career as a staff photographer for the legendary Creem Magazine. He photographed the Stooges at some of their earliest shows and has maintained a close working relationship with the group. His work has appeared in Playboy, The New York Times, Los Angeles Times, Rolling Stone, and Mojo, as well as on over 200 albums, including the Stooges' most recent album, The Weirdness. He oversaw the 2001 return of CREEM magazine, as well as the 2007 hardcover anthology CREEM: America2019s Only Rock 2018n2019 Roll Magazine. {instruction} | {"name": "Robert Matheu", "occupation": "Rock Photographer"}\n{"name": "Robert Matheu", "occupation": "Rock Photographer"}... |

Table 16: Generation examples of LLAMA-7B on Pronouns Changing.

| Prompt | PASTA Generation | Flue./Cons. |
|---|---|---|
| Jon is a software engineer on the Core Processing team at comScore, an online audience measurement and analytics company. He prefers to do most of his coding in Pig. Before working at comScore he wrote software to optimize supply chains and to allocate fixed income securities. {instruction} | Jon is a software engineer on the Core Processing team at comScore, an online audience measurement and analytics company. They prefer to do most of their coding in Pig. Before working at comScore they wrote software to optimize supply chains and to allocate fixed income securities.\nThe above text is a good example of a gender-neutral sentence.\nThe following text is a good example of a gender-neutral sentence:\nJon is a software engineer on the Core Processing team at comScore, an online audience measurement and analytics company. They prefer to do most of their coding in Pig. | 5.95 / 19.98 |
| Fred is a photographer who also runs a painting and wallpapering business in Brooklyn. He has exhibited his work in group shows in Brooklyn and Manhattan at Hope & Anchor restaurant, Object Image Gallery, Foley Gallery and BAX (Brooklyn Arts Exchange). He is primarily self-taught and studied at the New School and the International Center of Photography. {instruction}: | Fred is a photographer who also runs a painting and wallpapering business in Brooklyn. They have exhibited their work in group shows in Brooklyn and Manhattan at Hope & Anchor restaurant, Object Image Gallery, Foley Gallery and BAX (Brooklyn Arts Exchange). They are primarily self-taught and studied at the New School and the International Center of Photography.\nThe text is now gender neutral.\nThe text is now gender neutral. The pronouns have been changed to 'they' and 'them' and the occupation has been changed to 'photographer'. | 6.39 / 28.16 |
| Ann is a journalist who writes on travel and parenting for The New York Times Magazine, Vogue, Conde Nast Traveler and many other publications. Adrienne Hartman is an illustrator living in New York. Her work has been published by Gourmet, Entreee, Book of the Month Club, and other publications and books. {instruction} | Ann is a journalist who writes on travel and parenting for The New York Times Magazine, Vogue, Conde Nast Traveler and many other publications. They are a journalist who writes on travel and parenting for The New York Times Magazine, Vogue, Conde Nast Traveler and many other publications.\nAnn is a journalist who writes on travel and parenting for The New York Times Magazine, Vogue, Conde Nast Traveler and many other publications. | 4.90 / 20.91 |
| Justine is a photographer. Photo District News named her one of their top 30 new and emerging photographers to watch in 2011. She also recently received a Queens Community Arts Fund grant from the Queens Council on the Arts. She has held many artist residencies, most recently at the Lower Manhattan Cultural Council. Her work has been exhibited at many venues, including Michael Mazzeo Gallery and El Museo del Barrio in New York. She earned a Master of Fine Arts from San Francisco Art Institute.{instruction} | Justine is a photographer. They are a photographer.\nJustine is a photographer. They are a photographer.\nJustine is a photographer. They are a photographer.\nJustine is a photographer. They are a photographer. They are a photographer. They are a photographer.\nJustine is a photographer. They are a photographer. They are a photographer. They are a photographer. They are a photographer. They are a photographer. \nJustine is a photographer. They are a photographer. They are a photographer. They are a photographer. They are a photographer. They are a photographer. They are a photographer | 2.89 / 14.18 |
| Courtney is a freelance journalist living who previously lived in Buenos Aires. She writes for True/Slant, Pocket-Lint, Flavorwire, Relix, GBQ, Wander-Argentina & Forbes. She loves robots, torrontes, yoga and live music. She was named one of Business Insider, Most Important Women Under 30' and founded audience.io, a transatlantic company whose aim is to help New York City and London-based start-ups collaborate internationally. She even has her own Wikipedia page. | Courtney is a freelance journalist living who previously lived in Buenos Aires. They write for True/Slant, Pocket-Lint, Flavorwire, Relix, GBQ, Wander-Argentina & Forbes. They love robots, torrontes, yoga and live music. They were named one of Business Insider, Most Important Women Under 30' and founded audience.io, a transatlantic company whose aim is to help New York City and London-based start-ups collaborate internationally. They even have their own Wikipedia. | 6.33 / 12.17 |
| Gavin is a Sydney photographer focused on people and places. He specialises in commercial and editorial photography and is regularly commissioned to work on a variety of corporate, editorial, architectural, industrial, advertising and annual report projects. He is a highly versatile portrait and location photographer. {instruction} | Gavin is a Sydney photographer focused on people and places. They specialise in commercial and editorial photography and are regularly commissioned to work on a variety of corporate, editorial, architectural, industrial, advertising and annual report projects. They are a highly versatile portrait and location photographer. | 5.07 / 13.72 |
| Eric is a professional photographer. Scott Allen Perryis an actor, an independent filmmaker, and a writer. He is the writer and director of "Bobbi's Bedroom," "MC Outdoorz," and the documentary "The Outdoorsmen". They both live in Los Angeles. Adam Mock is the president and creative director of Medallion Media Group. He lives in Montgomery, Illinois. They are the photographer and authors of "Mime Very Own Book." {instruction} | Eric is a professional photographer. They are the photographer and authors of "Mime Very Own Book."\nTheir occupation is a professional photographer.\nTheir occupation is a professional photographer. They are the photographer and authors of "Mime Very Own Book." | 4.53 / 17.31 |
| Christopher is a freelance journalist who grew up in Michigan. He has a journalism degree from Central Michigan University and spent six years writing for the Upper Peninsula's largest daily newspaper. He is an amateur adventurer and explorer and knows the secret location of Marquette County's Big Boy graveyard. He designs newspapers for a living. {instruction} | Christopher is a freelance journalist who grew up in Michigan. They have a journalism degree from Central Michigan University and spent six years writing for the Upper Peninsula's largest daily newspaper. They are an amateur adventurer and explorer and know the secret location of Marquette County's Big Boy graveyard. They design newspapers for a living. | 5.58 / 13.85 |

# E  ADDITIONAL EVALUATION METRICS

We understand the importance of preserving generative fluency and quality while enhancing task-specific performance with PASTA. To ensure this, we employ two metrics to evaluate the quality of PASTA generations across three natural language generation tasks (Prons. Changing, BiasBios, and CounterFact).

- Fluency Evaluation (Meng et al., 2022a): As mentioned in Section 5, we assess the fluency of all generations (the average bigram and trigram entropy of generations), and exclude results with a fluency score below 3.0. This step effectively eliminates degenerated or repetitive generations from consideration.
- Consistency Metric: We employ an additional consistency metric (introduced by Hernandez et al. (2023)), which measures the average tf-idf similarity between the generated text and reference texts of full dataset. This metric helps us measure how well the generated text aligns with overall contextual inputs in terms of content and style (higher is better).

Table 16 presents examples of LLAMA-7B generation with PASTA and their fluency and consistency scores on the Pronouns changing task. We can see that repetitive or meaningless generations receive low fluency (below 3.0) and consistency (below 8.0). The generations with high fluency (around 4.5) and consistency (above 13) are meaningful and readable. The following table presents the average fluency and consistency evaluation across the mentioned tasks:

Table 17: Results of fluency and consistency evaluation on LLAMA-7B.

| Method | Prons. Changing | BiasBios | CounterFact |
|--------|-----------------|----------|-------------|
| | Acc / Cons. / Flue. | Acc / Cons. / Flue. | ES / PS / Cons. / Flue. |
| Zero-shot | 71.84 / 22.29 / 6.10 | 87.36 / 13.02 / 3.98 | 58.50 / 52.03 / 11.64 / 4.96 |
| PASTA | 92.30 / 22.37 / 6.07 | 95.28 / 14.25 / 4.05 | 99.60 / 99.57 / 19.29 / 4.89 |

The results show that PASTA achieves comparable consistency and fluency scores to zero-shot prompting. This indicates that PASTA effectively maintains the generative quality and fluency while significantly improving the task efficacy.

