# OpenReview forum: "Tell Your Model Where to Attend: Post-hoc Attention Steering for LLMs"
_ICLR.cc/2024/Conference — ICLR 2024 poster_

### Official Review · Reviewer_WBEq · 2023-10-30

**Soundness:** 1 poor
**Presentation:** 3 good
**Contribution:** 2 fair
**Rating:** 3
**Confidence:** 3

**Summary:**

This paper introduces a method to improve the quality of text generations on specific tasks by specifying explicitly to which portions of the prompt the model should pay close attention. To this end, the authors introduce 4 instruction tasks which have been annotated explicitly in order to place emphasis on a particular section of the prompt.

The authors run a "model profiling" phase on a training subset of each of these tasks. This phase aims at detecting which attention heads should see their outputs amplified in order for the LLM to follow more closely the operator's instructions. At inference time, the post-softmax attention scores of the corresponding heads are boosted by a constant factor \alpha.

The results show strong gains against raw LLMs in zero-shot and few-shot setups, and interesting robustness properties related to prompting.

**Strengths:**

* Originality: the proposed method for increasing attention weights for sections that have been manually emphasized is novel. The method is relatively lightweight, which makes it compelling to adapt very large models dynamically to new tasks.

* Quality: the code provided with the paper is problematic, as it means that the proposed evaluations may not correlate with human judgement on these tasks. In addition, a more thorough comparison with off-the-shelf instruction-tuned models may be relevant. See Weaknesses section for more details.

* Clarity: the paper is well-written and easy to follow. However, the definition of the main metrics could be updated to more closely match the code contents. See weaknesses section for more details.

* Significance: getting LLMs to adapt to new tasks at a reasonable cost is an exciting endeavour. However, human annotations are still required to emphasize specific chunks of texts. This human-in-the-loop requirement is comparable to that of instruction tuning, but it is not clear whether the results are comparable between the two methods.

**Weaknesses:**

# Problems in evaluation code

As can be seen in the code (https://anonymous.4open.science/r/PASTA-10E9/pasta/evaluator.py), the evaluation metrics for JSON Format and pronoun changing tasks do not seem aligned with any intuitive definition:

* the paper describes the metrics for the JSON format task as follows: "Format accuracy (F. Acc.) measures the accuracy at generating valid JSON. (b) Prediction accuracy (P. Acc.) measures the accuracy at generating the correct target in JSON values after loading the JSON-formatted generations.". However, in practice:
  * the evaluation for format accuracy measures whether there is any subsequence of the generated text that is valid json. It is highly debatable whether a model that would generate natural text intermingled with valid json is actually accurate at generating json.
  * the evaluation for prediction accuracy checks whether the desired string is present anywhere in the json values. As such, an answer that would print out e.g. all of the words in the context into one large json would score well.
* worse, the evaluation for pronoun change looks only at whether specific pronouns ("she", "he", "her"...) are no longer in the generated text, without verifying that the rest of the text is unchanged. As such, an empty generation "" would score 100. This explains partially Figure 3b, where the model's fluency seems to be decreasing steadily as more heads are steered, while the pronoun change accuracy keeps increasing.

These flaws in evaluation are troubling: a high score on these metrics may be paired with a substantial decrease in overall fluency and perceived quality of the model. This does not seem in line with the paper's stated contribution which is to "enhance an LLM’s ability to follow user instructions".

# Overlap with instruction-tuning

The main focus of the paper is to get models to generate text that follows specific instructions. While the authors cite existing publications on instruction-tuning, their claim that "PASTA can be used in addition to these approaches to improve some aspects of model steerability" would need to be substantiated. Indeed, one could assume that instruction fine-tuning will alter attention weights to have the model attend closely to the operator's instructions. The notion that PASTA improvements would transfer to such models would thus need to be tested. Furthermore, the cost of the model profiling phase should be compared to the cost of a lightweight instruction-tuning setup.

**Questions:**

* Could you clarify how we should interpret results on json generation and pronoun change given the shortcomings of the evaluation code?

* Do the findings transfer on readily-available instruction-tuned models?

* In effect, the prompt the paper is operating on has two components: the input text and the emphasis. In section 5.2, you demonstrate that PASTA mitigates noise and poor wording choices in the input test. For a full comparison with zero-shot prompting, could you test the effect of poor choice of emphasis in the input text?

* In the same section, what is the reason to compare against zero-shot and not a few-shot setup? It would seem that few shot prompting would be a fairer comparison against multi-task PASTA, as both assume that they have access to some training examples on the task at hand.

---

> ### Author Response · Authors · 2023-11-21
> **Thanks for your constructive feedback.**
>
> Thanks for your constructive feedback. We provide our response to your questions as follows.
>
> **Q1: The evaluation measures whether there is any subsequence of the generated text that is valid json. It is highly debatable whether a model that would generate natural text intermingled with valid json is actually accurate at generating json.**
>
> Thanks for your feedback regarding our evaluation. To clarify, our decision to evaluate subsequences as valid JSON strings stems from the model's behavior during generation. We observed that when the model successfully follows the instruction to generate a JSON string, it tends to repeat the correct JSON strings until reaching the maximum token limit. For instance, an output might look like ‘{"name":"Mary", "occupation":"doctor"} {"name":"Mary", ...’. Moreover, models occasionally preface JSON strings with natural language text, such as ‘The answer is {“name”, “Mary”, …’. To accommodate these variations and fairly assess the model's capability, we chose to evaluate whether any subsequence, starting from '{' and ending with '}', forms a valid JSON string.
>
> To provide a comprehensive view of the model's performance, we have included all generation results by LLAMA-7B on the JSON Formatting task in the link of https://anonymous.4open.science/r/PASTA-10E9/model_generations/json_formatting/llama7b_pasta_generation.json (the table blew shows a few examples). We invite the reviewer to examine these results in detail. These generation results showcase the practical effectiveness of the PASTA approach in guiding the model to generate valid JSON structures.
>
> | Prompt | PASTA Generation |
> | :-----: | :-----: |
> | David is an attorney in San Antonio. He doesn’t dress up for work and he doesn’t shred his client files. He was caught throwing personal information…{instruction} | {"name": "David", "occupation": "attorney"}\n{"name": "David", "occupation": "attorney"}... |
>  | Juris is an associate professor at the Faculty of Information, University of Toronto. He is a co-author of Readers' Advisory Service in North American…{instruction} | {"name": "Juris", "occupation": "Associate Professor"}\n{"name": "Juris", "occupation": "Associate Professor"}... |
> | Gregory is an educational psychologist whose major concern is with how people learn, ie skills and knowledge development especially in…{instruction} | {"name": "Gregory", "occupation": "Educational Psychologist"}\n{"name": "Gregory", "occupation": "Educational Psychologist"}...|
>
> **Q2: The prediction accuracy checks if the desired string is present anywhere in the json values. As such, an answer that would print out e.g. all of the words in the context into one large json would score well.**
>
> In the biographical data, there are 28 target labels, each of which is a short string, e.g. professor, teacher, nurse. In practical generations, models tend to generate JSON values that contain more detailed information derived from the context, such as “assistant professor of computer science”, “English teacher”, “retired nurse”, etc (see examples in the table of Q1). To accommodate these variations given different contexts, we evaluate the prediction accuracy by checking if the short string of labels is present in the JSON values.
>
> To further ensure a rigorous evaluation, we reevaluate the results with an additional criterion: the JSON values must be concise, with a word limit (e.g., 10 words for ‘professor’ class and 3 words for the others). A re-evaluation of the prediction accuracy with the word limit criterion yielded an accuracy of 84.42% for LLAMA-7B using PASTA, only marginally lower than the originally reported 85.09% in Table 1. This minimal discrepancy can support our claim that PASTA effectively guides the model to generate relevant and concise JSON values. The detailed generation results, available at https://anonymous.4open.science/r/PASTA-10E9/model_generations/json_formatting/llama7b_pasta_generation.json , demonstrate that the occurrence of the model outputting entire contexts as JSON values is much infrequent.
>
> We believe these measures robustly evaluate the model's prediction accuracy and effectively mitigate the concern of scoring well through irrelevant verbosity. We invite the reviewer to review the provided results to further assess the effectiveness of our approach.

---

> ### Author Response · Authors · 2023-11-21
> **The response to the questions about Pronouns changing task**
>
> **Q3: The evaluation for pronoun change looks only at whether specific pronouns ("she", "he", "her"...) are no longer in the generated text, without verifying that the rest of the text is unchanged. As such, an empty generation would score 100.**
>
> Thanks for your comments. We understand the importance of not only changing pronouns but also maintaining textual integrity and generation quality. To ensure this, we employ two evaluation metrics:
>
> 1. Fluency Evaluation (Meng et al., 2022): As mentioned in Section 4, we assess the fluency of all generations (the average bigram and trigram entropy of generations), and exclude results with a fluency score below 3.0. This step effectively eliminates degenerated or repetitive generations from consideration, such as empty text.
>
> 2. Consistency Metric: We employ an additional consistency metric (introduced by Hernandez et al., (2023)), which measures the average tf-idf similarity between the generated text and reference texts of full dataset. This metric helps us measure how well the generated text aligns with overall contextual inputs in terms of content and style.
>
> The table below presents a few examples of LLAMA-7B generation with PASTA and their fluency and consistency scores (please see more in Appendix D and all generation results are available at https://anonymous.4open.science/r/PASTA-10E9/model_generations/pron_changing/llama7b_pasta_generation.json).
>
> | Prompt | PASTA generation | Flue. / Cons. |
> | :-----: | :-----: | :-----: |
> | Sawako is a teacher at Sakura High. She is originally the adviser of the wind instrument club but then she was forced to become the adviser of the light music club…{instruction} | Sawako is a teacher at Sakura High. They are originally the adviser of the wind instrument club but then they were forced to become the adviser of the light music club… | 5.21 / 17.53 |
> | Catherine is a Sydney-based journalist. She has written for both Fairfax and News Ltd. She has never listened to Alan Jones' radio show…{instruction} | Catherine is a Sydney-based journalist. They have written for both Fairfax and News Ltd. They have never listened to Alan Jones' radio show… | 4.91 / 15.30 |
> | Cynthia is an assistant professor of History at UND. Her book, Gender and Generation on the Far Western Frontier, traces changes in…{instruction} | Cynthia is a historian who studies the history of the American West. They are currently working on a book about the history of the American West… | 4.51 / 23.24 |
> | Shahid is an immigration attorney in Helena, Montana and represents…{instruction} | Shahid is a they who is a they who is a they who is a they… | 2.05 / 5.56 |
>
> From the examples, we can see that repetitive or meaningless generations receive low fluency (below 3.0) and consistency (below 8.0). The generations with high fluency (around 5.0) and consistency (above 13) are meaningful and readable. The following table presents the results of average consistency and fluency evaluation on the Pronouns changing task.
>
> | **LLAMA-7B** | Acc |  **Consistency** | **Fluency** |
> | :-----: | :-----: | :-----: | :-----: |
> | Zero-shot | 71.84 |  22.29 | 6.10 |
> | *-marked | 39.14 | 22.20 | 6.03 |
> | “”-marked | 20.55 | 13.17 | 5.13 |
> | Few-shot | 59.06 | 23.46 |  5.95 |
> | PASTA($\|\mathcal{H}\|=86$) | 96.42 | 19.03 | 5.43 |
> | PASTA($\|\mathcal{H}\|=17$) | 92.30 | 22.37 | 6.07 |
>
>
> | **GPT-J** | Acc | **Consistency** | **Fluency** |
> | :-----: | :-----: | :-----: | :-----: |
> | Zero-shot | 39.88 | 20.12 | 5.91 |
> | *-marked | 41.25 | 13.71 | 4.76 |
> | “”-marked | 6.12 | 13.89 | 5.43 |
> | Few-shot | 35.77 | 22.12 | 6.46 |
> | PASTA | 92.96 | 17.98 | 4.91 |
>
> These results indicate that PASTA achieves comparable consistency and fluency scores to other prompting methods, suggesting that it effectively maintains the quality of generated texts. The slightly lower consistency score of PASTA than zero-shot prompting is a result of the necessary pronoun substitutions. We provide all generation results in the link above which show how PASTA performs in maintaining text integrity while effectively implementing pronoun changes.
>
> **Q4: Clarification of the results in Figure 2b**
>
> Figure 2b indicates the drawbacks of overemphasizing. When users overemphasize specific parts by steering too many heads (e.g., 294), it leads the model to only focus on the specific information while ignoring other parts. This imbalance results in the deterioration of overall generation quality. In the example of Pronouns changing task, when 294 heads are steered, the model becomes overly fixated on emphasized parts, leading to repeated generations, such as “Lindsey is a textile worker. They are a textile worker. They are …”, This deterioration reflected in the decreased consistency score of 7.09 and fluency score of 3.16. Therefore, Figure 2b suggests applying PASTA to a moderate number of heads (typically 30 to 150) is ideal, which strikes a balance between effective emphasis and generation quality.

---

> ### Author Response · Authors · 2023-11-21
> **The transferability of PASTA to instruction-tuned models**
>
> **Q5: The notion that PASTA improvements would transfer to such models would thus need to be tested.**
>
> We test PASTA’s applicability to Vicuna-7B-v1.3, an instruction-tuned model from LLAMA-7B.  We apply PASTA using attention heads selected from LLAMA-7B profiling, including intersection and task-specific heads. Then we evaluate the performance of Vicuna across four tasks. The table below presents the results.
>
> | Models | JSON. F. Acc | JSON Pred. Acc | Prons. Acc. | Prons. Consistency | BiasBios Acc | CounterFact ES | CounterFact PS |
> | :-----: | :-----: | :-----: | :-----: | :-----: | :-----: | :-----: | :-----: |
> | LLAMA Zero-shot | 60.00 | 54.94 | 71.84 | 22.29 | 87.36 | 58.50 | 52.03 |
> | LLAMA PASTA(intersection) | 96.64 | 85.09 | 96.42 | 21.91 | 95.28 | 99.60 | 99.57 |
> | Vicuna Zero-shot | 65.41 | 61.78 | 95.74 | 21.83 | 90.74 | 61.10 | 52.46 |
> | Vicuna PASTA(intersection) | 66.09 | 56.00 | 98.82 | 22.59 | 96.44 | 99.80 | 99.80 |
> | Vicuna PASTA(task-specific) | 90.54 | 86.53 | 98.62 | 22.48 | 97.42 | 99.82 | 99.74 |
>
> The results demonstrate that the attention heads selected for LLAMA-7B effectively steer Vicuna-7B, indicating that re-profiling is not necessary for instruction-tuned models. Notably, when steering task-specific heads selected from LLAMA profiling, PASTA significantly enhances Vicuna's performance across all tasks. This evidence shows that PASTA can complement instruction tuning to further improve the model ability of instruction following.
>
> **Q6: Furthermore, the cost of the model profiling phase should be compared to the cost of a lightweight instruction-tuning setup.**
>
> Profiling for PASTA is an inference-only process, conducted on a small subset of the training set. For instance, profiling all heads of LLAMA-7B with 1000 samples requires 1000 * 32 * 32 inferences. In contrast, fine-tuning a model like Vicuna-7B on a dataset such as 128k ShareGPT data for 8 epochs demands an equal number of forward passes, along with additional backward passes that incur much higher memory footprints. Fine-tuning also involves multiple trials of hyperparameter tuning, whereas profiling is performed only once for each model. Moreover, fine-tuning requires maintaining separate copies of fine-tuned model weights while PASTA does not incur any additional model checkpoints and can be applied plug-and-play.
>
> Meanwhile, the robustness of head performance ranking to sample variance allows us to further reduce the sample size for profiling (e.g., $|\mathcal{D}|=200$). The new table below presents the PASTA performance on the JSON Formatting task when re-profiling with $|\mathcal{D}|=200$ samples. We can see PASTA still achieves superior performance when profiling with much fewer examples.
>
> | Model | Profiling Sample size | JSON F. Acc | JSON Pred. Acc |
> | :-----: | :-----: | :-----: | :-----: |
> | LLAMA Zero-shot | N/A | 60.00 | 54.94 |
> | LLAMA w. PASTA | 1000 | 95.56 | 86.83 |
> | LLAMA w. PASTA | 200 | 95.04 | 84.17 |
> | Vicuna Zero-shot | N/A | 65.41 | 61.78 |
> | Vicuna w. PASTA | 200 | 98.56 | 94.56 |

---

> ### Author Response · Authors · 2023-11-21
> **Author response to the questions about Section 5.2**
>
> **Q7: For a full comparison with zero-shot prompting, could you test the effect of poor choice of emphasis in the input text?**
>
> We would like to remind the reviewer that PASTA allows users to emphasize any part of the input text to guide model generation. However, the effectiveness of this emphasis largely depends on capturing the essential information intended to be highlighted. In Section 5.2, we show that PASTA consistently improves instruction following when emphasizing the entire instruction, regardless of how they are rephrased. We can only highlight instructions here to illustrate this. To further explore the varying emphasis, we conducted an additional ablation to test the effectiveness of PASTA when only emphasizing partial information of instructions on the JSON Formatting task. The instruction of this task consists of two parts: (i) (question) “Answer the occupation of {person} and generate the answer as json format”; (ii) (demonstration) “Here is an example: {“name”: , “occupation”: ,}”. We evaluate the performance of PASTA when emphasizing only partial information (question (i) or demonstration (ii)):
>
> | Method | JSON F. Acc | JSON P. Acc |
> | :-----: | :-----: | :-----: |
> | Zero-shot | 60.00 | 54.94 |
> | PASTA emphasizes the whole instruction | 96.64 | 85.09 |
> | PASTA only partially emphasizes (i) | 96.14 | 85.75 |
> | PASTA only partially emphasizes (ii) | 82.29 | 72.79 |
> | PASTA emphasizes other parts except for instruction | 0.88 | 0.83 |
>
> The results from the table above indicate that while full emphasis on the instruction yields the best performance, PASTA still performs effectively even when only a part of the instruction is emphasized, indicating the robustness of PASTA on varying emphasis. Moreover, we can have models overlook the instruction entirely by emphasizing the other parts (contexts) except for instruction. In this way, the model is biased to totally ignore the instruction, leading to almost zero-scored performance. Hence, these results reveal that an effective emphasis should be accurate and informative enough to reflect the user's intention.
>
> **Q8: In Section 5.2, what is the reason to compare against zero-shot and not a few-shot setup?**
>
> We mainly present zero-shot performance as the baseline in Section 5.2 as this section specifically focuses on the model's sensitivity to prompt rephrasing in our tasks, which is clearer in the zero-shot setting. The table below includes the performance of few-shot prompting of LLAMA-7B under the same settings.
>
> | Instruction | Method | JSON F. Acc / P. Acc | Prons. Acc / A. Acc |
> | :-----: | :-----: | :-----: | :-----: |
> | Original | Zero-shot | 60.0 / 54.9 | 71.8 / 66.3 |
> | Original | Few-shot | 84.9 / 73.6 | 59.1 / 55.3 |
> | Original | PASTA | 96.6 / 85.1 | 96.4 / 95.8 |
> | Shortened | Zero-shot | 36.0 / 32.4 | 49.2 / 42.6 |
> | Shortened | Few-shot | 95.9 / 81.53 | 57.6 / 51.4 |
> | Shortened | PASTA | 87.4 / 65.9 | 89.0 / 86.9 |
> | Rephrased | Zero-shot | 57.9 / 54.2 | 82.3 / 79.6 |
> | Rephrased | Few-shot | 86.7 /  75.3 | 86.7 / 84.1 |
> | Rephrased | PASTA | 97.1 / 87.1 | 89.6 / 89.0 |

---

> ### Comment · Reviewer_WBEq · 2023-11-22
>
> I thank the authors for their responses. While the additional data they provide is reassuring, overall I am still concerned by the brittleness of the proposed evaluation metrics:
> 1. while it is good to know that altered models hold a similar bigram and trigram distribution (see definition of fluency), there is a wide gap between this and an answer that is semantically close to the desired outcome.
> 2. The accuracy metric defined in the paper does not reflect the code implementation: "Accuracy evaluates the ratio that ‘she/he’ are successfully changed to ‘they’ in model generations." However, as noted in my first review, the accuracy metric defined in the code measures how accurate the model is at *removing* said pronouns, not at changing them.
> 3. The Json accuracy metrics implementations also do not match their definition in the paper. "Prediction accuracy (P. Acc.) measures the accuracy at generating the correct target in JSON values after loading the JSON-formatted generations." implies that the model (1) generates only one Json output and (2) that the desired output is unique.
>
>
> As these discrepancies are meaningful to the paper and have not been addressed in the comments, I do not wish to change my rating for this paper.

---

### Official Review · Reviewer_pBs7 · 2023-10-31

**Soundness:** 2 fair
**Presentation:** 3 good
**Contribution:** 3 good
**Rating:** 6
**Confidence:** 4

**Summary:**

This paper proposes a post-hoc method, called PASTA, that allows users to steer the model's attention towards user-specified information by highlighting relevant parts of the input. The idea is to select a small subset of attention heads and applies precise attention reweighting on them; that is, we select some attention heads and steer the corresponding attention weight to focuses more on the specified input spans. This proposed method is evaluated on four tasks with GPT-J-6B and LLaMA-7B. Experimental results show that PASTA achieves an average accuracy improvement.

**Strengths:**

This paper is easy to read and the method is not difficult to implement. The idea of steering attention to enhance large language models' adherence to instructions proffers significant value. This paper also introduces a PASTA method, which can select a small subset of attention heads and applies precise attention reweighting on them during the inference process.

**Weaknesses:**

- This proposed method necessitates the predefined highlighted input spans. However, in the context of numerous tasks, defining such input spans to be a formidable challenge. One idea is whether an automatic extraction mechanism could be designed to improve this method. This could entail the development of a dedicated extraction model, or the crafting of specialized prompts to guide Large Language Models in executing this task more effectively.
- The PASTA method does not necessitate alterations in model parameters, resulting in enhanced performance efficiency; however, a pertinent issue arises concerning the potential disruption of the established generative law from language modeling via manual attention steering. This makes a conjecture that such intervention could inadvertently impair other abilities of LLMs, such as generative fluency and reasoning.
- The current experimental design appears to be limited in scope. For a more comprehensive evaluation, it would be beneficial to test the effectiveness of the proposed method on diverse large language models, including but not limited to, 7B-LoRA, 13B-LoRA, and 13B, across specific tasks.
- There have been previous studies on improving the standard attention with the prior knowledge, such as syntax-based attention and multiscale attention. These should be considered baselines for comparison, either in related work or experiments. Additionally, although the four tasks previously considered do not adequately represent the breadth of generative tasks commonly utilized as benchmarks, they do not represent the prevailing benchmarks within generative tasks. It is crucial to evaluate the proposed methodology's efficacy by applying it to some commonly used generative tasks, including but not limited to, machine translation and question answering tasks.

**Questions:**

- Did you perform the SFT produce on the models in the experiment?
- You should present more details in computing accuracy on these used datasets.

---

> ### Author Response · Authors · 2023-11-21
> **We appreciate the reviewer for the constructive feedback.**
>
> We appreciate the reviewer for your constructive feedback. We would like to provide our response to your questions as follows.
>
> **Q2: This makes a conjecture that such intervention could inadvertently impair other abilities of LLMs, such as generative fluency and reasoning.**
>
> Thanks for your comment. We fully understand the importance of preserving generative fluency and quality while enhancing task-specific performance with PASTA. To ensure this, we employ two metrics to evaluate the quality of PASTA generations across three natural language generation tasks (Prons. Changing, BiasBios, and CounterFact).
>
> 1. Fluency Evaluation (Meng et al., 2022): As mentioned in Section 4, we assess the fluency of all generations (the average bigram and trigram entropy of generations), and exclude results with a fluency score below 3.0. This step effectively eliminates degenerated or repetitive generations from consideration.
>
> 2. Consistency Metric: We employ an additional consistency metric (introduced by Hernandez et al., (2023)), which measures the average tf-idf similarity between the generated text and reference texts of full dataset. This metric helps us measure how well the generated text aligns with overall contextual inputs in terms of content and style (higher is better).
>
> The table below presents a few examples of LLAMA-7B generation with PASTA and their fluency and consistency scores on the Pronouns changing task (please see Appendix D for more examples).
>
> | Prompt | PASTA generation | Flue. / Cons. |
> | :-----: | :-----: | :-----: |
> | Sawako is a teacher at Sakura High. She is originally the adviser of the wind instrument club but then she was forced to become the adviser of the light music club…{instruction} | Sawako is a teacher at Sakura High. They are originally the adviser of the wind instrument club but then they were forced to become the adviser of the light music club… | 5.21 / 17.53 |
> | Catherine is a Sydney-based journalist. She has written for both Fairfax and News Ltd. She has never listened to Alan Jones' radio show…{instruction} | Catherine is a Sydney-based journalist. They have written for both Fairfax and News Ltd. They have never listened to Alan Jones' radio show… | 4.91 / 15.30 |
> | Cynthia is an assistant professor of History at UND. Her book, Gender and Generation on the Far Western Frontier, traces changes in gender roles…{instruction} | Cynthia is a historian who studies the history of the American West. They are currently working on a book about the history of the American West… | 4.51 / 23.24 |
> | Shahid is an immigration attorney in Helena, Montana and represents…{instruction} | Shahid is a they who is a they who is a they who is a they… | 2.05 / 5.56 |
>
> From the examples, we can see that repetitive or meaningless generations receive low fluency (below 3.0) and consistency (below 8.0). The generations with high fluency (around 4.5) and consistency (above 13) are meaningful and readable. The following table presents the average fluency and consistency evaluation across the mentioned tasks:
>
> | Task | Metric | Zero-shot| PASTA |
> | :-----: | :-----: | :-----: | :-----: |
> | Prons. Changing | Acc | 71.84 | 92.30 |
> | Prons. Changing | Consistency | 22.29 | 22.37 |
> | Prons. Changing | Fluency | 6.10 | 6.07 |
> | BiasBios | Acc | 87.36 | 95.28 |
> | BiasBios | Consistency | 13.02 | 14.25 |
> | BiasBios | Fluency | 3.98 | 4.05 |
> | CounterFact | Efficacy score | 58.50 | 99.60 |
> | CounterFact | Paraphrase score | 52.03 | 99.57 |
> | CounterFact | Consistency | 11.64 | 19.29 |
> | CounterFact | Fluency | 4.96 | 4.89 |
>
> The results show that PASTA achieves comparable consistency and fluency scores to zero-shot prompting. This indicates that PASTA effectively maintains the generative quality and fluency while significantly improving the task efficacy.
>
> Meanwhile, we would like to kindly remind the reviewer that Section 5.3 (Figures 3a and 3b) has discussed the reviewer’s conjecture. The generative fluency can be impacted only when users overemphasize highlighted input spans by steering too many heads (e.g., 294). Such excessive emphasis can cause the model to focus narrowly on specific information while ignoring other parts of inputs, decreasing consistency and fluency. Therefore, we recommend applying PASTA to a moderate number of heads: typically between 30 to 150. Steering this range of heads already yields significant improvement in task performance while maintaining high generative quality, striking a good balance between emphasis efficacy and generative fluency.

---

> ### Author Response · Authors · 2023-11-21
> **More evaluation results**
>
> **Q1: Could an automatic extraction mechanism be designed to improve this method? This could entail the development of a dedicated extraction model or the crafting of specialized prompts.**
>
> We appreciate the reviewer’s comment on potential exploration for improving PASTA. Your suggestion of developing an automatic extraction mechanism is indeed a valuable direction for future work. The primary contribution of PASTA lies in empowering users to selectively highlight information according to their intentions, thereby effectively steering model generation. As an initial work in post-hoc attention steering, we motivate future efforts to combine PASTA with methods capable of automatically detecting input spans relevant to user intentions. For example, one potential approach is to employ a syntax parser to dissect the structure of input strings and analyze the sensitivity of each part to model outputs. The ideas suggested by the reviewer are also interesting – tailoring an auxiliary model or specializing prompts to extract the task-specific emphases for PASTA. We believe these possibilities can further promote the development of attention steering, offering more effective ways to guide model generations.
>
> **Q3: it would be beneficial to test the effectiveness of the proposed method on diverse large language models, including but not limited to, 7B-LoRA, 13B-LoRA, and 13B, across specific tasks.**
>
> We conduct experiments with LLAMA-13B and Vicuna-7B to further evaluate the effectiveness of PASTA across all tasks. The following table presents the performance comparison for LLAMA-13B (Flues./Cons. means fluency/consistency scores).
>
> | Method | JSON Formatting | Prons. Changing | Prons. Changing | BiasBios | BiasBios | CounterFact | CounterFact |
> | :-----: | :-----: | :-----: | :-----: | :-----: | :-----: | :-----: | :-----: |
> |  | F. / P. Acc | Acc / A. Acc | Flue./Cons. | Acc | Flue./Cons. | ES / PS | Flue./Cons. |
> | Zero-shot | 45.48/43.16 | 65.03/60.90 | 6.11/22.50 | 85.80 | 4.52/13.89 | 47.86/44.14 | 5.06/11.10 |
> | Few-shot | 39.80/3.56 | 82.33/80.71 | 5.19/22.44 | 88.38 | 4.91/18.04 | 90.63/65.49 | 5.93/11.13 |
> | PASTA (multi-task) | 98.74/89.88 | 97.56/96.78 | 6.07/22.95 | 95.34 | 4.61/15.72 | 99.38/99.30 | 5.07/17.04 |
>
> The results demonstrate that PASTA significantly improves the performance of LLAMA-13B across all tasks, indicating its generability to larger model sizes. Meanwhile, PASTA maintains or improves fluency and consistency scores compared to zero-shot prompting on three of the tasks, suggesting that PASTA does not impair model generative ability and fluency.
>
> We further test PASTA’s applicability to Vicuna-7B-v1.3, which is instruction-tuned from LLAMA-7B. We apply PASTA using attention heads selected from LLAMA-7B profiling (including multi-task and task-specific heads). In this way, we evaluate if the heads selected from the base models are transferable to an instruction-tuned model, thereby avoiding the re-profiling. The table below presents the performance of Vicuna across all tasks.
>
> | Method | JSON Formatting | Prons. Changing | Prons. Changing | BiasBios | BiasBios | CounterFact | CounterFact |
> | :-----: | :-----: | :-----: | :-----: | :-----: | :-----: | :-----: | :-----: |
> |  | F. / P. Acc | Acc / A. Acc | Flue./Cons. | Acc | Flue./Cons. | ES / PS | Flue./Cons. |
> | LLAMA Zero-shot | 60.00/54.94 | 71.84/66.28 | 6.10/22.29 | 87.36 | 3.98/13.02 | 58.50/52.03 | 4.96/11.64 |
> | LLAMA PASTA(multi-task) | 96.64/85.09 | 96.42/95.84 | 5.43/21.91 | 95.28 | 4.05/14.25 | 99.60/99.57 | 4.89/19.29 |
> | Vicuna Zero-shot | 65.41/61.78 | 95.74/94.74 | 5.75/21.83 | 90.74 | 4.20/16.00 | 61.10/52.46 | 5.07/12.76 |
> | Vicuna PASTA(multi-task) | 66.09/56.00 | 98.82/98.08 | 5.87/22.59 | 96.44 | 4.60/16.79 | 99.80/99.80 | 5.11/21.26 |
> | Vicuna PASTA(task-specific) | 90.54/86.53 | 98.62/98.04 | 5.81/22.48 | 97.42 | 4.56/16.24 | 99.82/99.74 | 5.03/20.28 |
>
> The results demonstrate that the attention heads selected for LLAMA-7B effectively steer Vicuna-7B, indicating that re-profiling is not necessary for instruction-tuned models. Notably, when steering task-specific heads selected from LLAMA profiling, PASTA significantly enhances Vicuna's performance across all tasks. This evidence shows that PASTA can complement instruction tuning without necessitating re-profiling.

---

> ### Author Response · Authors · 2023-11-21
> **The response to the other questions**
>
> **Q4: The previous studies on syntax-based attention and multiscale attention should be considered baselines for comparison, either in related work or experiments.**
>
> Thanks for your feedback regarding the relevance of prior studies on syntax-based and multiscale attention. Multi-scale attention [3] introduces varied attention scales across different heads, enabling diverse functionalities. Syntax-based attention [4,5], leverages the syntactic structures of inputs to modify attention patterns. However, these attention mechanisms require additional training to adapt to existing LLMs like LLAMA. By contrast, PASTA does not require additional training or alternations in model parameters, which allows PASTA to be seamlessly integrated with existing LLMs. We appreciate your comments on these insightful prior works and will discuss them in the related work of our next version.
>
> **Q5: It is crucial to evaluate the proposed methodology's efficacy by applying it to some commonly used generative tasks, such as machine translation and question-answering tasks.**
>
> Thanks for your valuable feedback regarding more experimental comparisons on commonly used tasks, such as MT and QA. We fully recognize the importance of evaluating our methodology across these standard benchmarks and make every effort to conduct these additional comparisons during the rebuttal phase. However, we cannot access the same computational resources as before. Due to the limited computational capacity and time, we may not be able to finish all these additional experiments on time. Please be assured that we are committed to expanding the scope of our research. We will prioritize completing these experiments and will share the new results as soon as they are available.
>
> **Q6: Response to other questions.**
>
> Regarding the question of supervised fine-tuning (SFT), PASTA is an inference-only method and does require any fine-tuning or re-training. Therefore, we did not perform any fine-tuning for the models. All evaluations are performed on the pre-trained models.
> We will include detailed implementations of accuracy evaluation in the appendix of our next version.
>
> [1] Kevin Meng, David Bau, Alex Andonian, and Yonatan Belinkov. Locating and editing factual associations in gpt. Advances in Neural Information Processing Systems, 35:17359–17372, 2022.
>
> [2] Evan Hernandez, Belinda Z. Li, and Jacob Andreas. Inspecting and editing knowledge representations in language models, 2023.
>
> [3] Qipeng Guo, Xipeng Qiu, Pengfei Liu, Xiangyang Xue, Zheng Zhang. Multi-Scale Self-Attention for Text Classification. AAAI 2020.
>
> [4] PengFei Liu, Xipeng Qiu, Xuanjing Huang. Syntax-Based Attention Masking for Neural Machine Translation. 2016.
>
> [5] Colin McDonald and David Chiang. Syntax-Based Attention Masking for Neural Machine Translation. NAACL 2021.

---

> > ### Comment · Reviewer_pBs7 · 2023-11-22
> > **After Rebuttal**
> >
> > I would like to extend my sincere appreciation to the authors for their efforts in addressing the concerns raised during the rebuttal process. The displayed new results, including the comparison in terms of consistency and fluency, experiments on LLAMA and Vicuna, make the current manuscript more convincing. I am pleased to note that most of my concerns have been adequately addressed. Thus I raised the score to 6.

---

### Official Review · Reviewer_s9kK · 2023-10-31

**Soundness:** 3 good
**Presentation:** 4 excellent
**Contribution:** 3 good
**Rating:** 8
**Confidence:** 4

**Summary:**

This paper introduces PASTA (Post-hoc Attention STeering Approach), which addresses the need to guide the attention of large language models (LLMs) towards user-specified information. Unlike existing methods that are limited to plain text processing, PASTA enables LLMs to read text with user-specified emphasis marks, mimicking the subtleties of text style in human-written articles. PASTA achieves this by identifying a select group of attention heads and applying precise attention reweighting, directing the model's focus towards user-specified parts of the text. The method is applied at inference time and does not require any changes to the model parameters. Experimental results demonstrate that PASTA significantly improves an LLM's ability to follow user instructions and integrate new knowledge from user inputs. The performance improvement is substantial across various tasks, with an average accuracy boost of 22% observed for LLAMA-7B.

**Strengths:**

1.	The authors propose a simple yet effective method called post-hoc attention steering. The method improves performance in multi-task and task-agnostic settings by forcing the model to focus on several important positions.

2.	The structure of this paper is really easy to follow and understand.

3.	The experiments are pretty comprehensive from my perspective, all the claims are properly proved.

**Weaknesses:**

1.	Experiments on larger models should be conducted to ensure the performance across different models.

**Questions:**

1.	One thing I am really curious about is whether we have any preferences layer-wise. Since different layers are known to possess different functions. Which layer is more important for attention steering? By Figure 2 it seems that it’s approximately random, is there any explanation for this?

---

> ### Author Response · Authors · 2023-11-21
> **We appreciate the reviewer for the valuable feedback.**
>
> We appreciate the reviewer for recognizing the novelty and effectiveness of our methods. We would like to provide our response to the reviewer’s question as follows.
>
> **Q1: Experiments on larger models should be conducted to ensure the performance across different models.**
>
> We conduct experiments with LLAMA-13B and Vicuna-7B to further evaluate the effectiveness of PASTA across all tasks. The following table presents the performance comparison for LLAMA-13B.
>
> | Method | JSON Formatting | Prons. Changing | BiasBios | CounterFact |
> | :-----: | :-----: | :-----: | :-----: | :-----: |
> |  | F. / P. Acc | Acc / A. Acc | Acc | ES / PS |
> | Zero-shot | 45.48/43.16 | 65.03/60.90 | 85.80 | 47.86/44.14 |
> | Few-shot | 39.80/3.56 | 82.33/80.71 | 88.38 | 90.63/65.49 |
> | PASTA (multi-task) | 98.74/89.88 | 97.56/96.78 | 95.34 | 99.38/99.30 |
>
> The results demonstrate that PASTA significantly improves the performance of LLAMA-13B across all tasks, indicating its generability to larger model sizes.
>
> We further test PASTA’s applicability to Vicuna-7B-v1.3, which is instruction-tuned from LLAMA-7B. We apply PASTA using attention heads selected from LLAMA-7B profiling (including multi-task and task-specific heads). In this way, we evaluate if the heads selected from the base models are transferable to an instruction-tuned model, thereby avoiding the re-profiling. The table below presents the performance of Vicuna across all tasks.
>
> | Method | JSON Formatting | Prons. Changing |  BiasBios | CounterFact |
> | :-----: | :-----: | :-----: | :-----: | :-----: |
> |  | F. / P. Acc | Acc / A. Acc | Flue./Cons. | Acc | Flue./Cons. | ES / PS | Flue./Cons. |
> | LLAMA Zero-shot | 60.00/54.94 | 71.84/66.28 | 87.36 | 58.50/52.03 |
> | LLAMA PASTA(multi-task) | 96.64/85.09 | 96.42/95.84 | 95.28 | 99.60/99.57 |
> | Vicuna Zero-shot | 65.41/61.78 | 95.74/94.74 | 90.74 | 61.10/52.46 |
> | Vicuna PASTA(multi-task) | 66.09/56.00 | 98.82/98.08 | 96.44 | 99.80/99.80 |
> | Vicuna PASTA(task-specific) | 90.54/86.53 | 98.62/98.04 | 97.42 | 99.82/99.74 |
>
> The results demonstrate that the attention heads selected for LLAMA-7B effectively steer Vicuna-7B, indicating that re-profiling is not necessary for instruction-tuned models. Notably, when steering task-specific heads selected from LLAMA profiling, PASTA significantly enhances Vicuna's performance across all tasks. This evidence shows that PASTA can complement instruction tuning without necessitating re-profiling.
>
> **Q2: Since different layers are known to possess different functions. Which layer is more important for attention steering? By Figure 2 it seems that it’s approximately random, is there any explanation for this?**
>
> Thanks for your great comment. According to our observation (please see Figures 2,4,5,6), there are some layers like layer 19 in LLAMA-7B that can be generally steered for all of the tasks to improve the performance. These layers can be applied to attention steering in task-agnostic manners. In most cases, each task prefers different layers for attention steering, which should be used task-specifically. More fine-grainedly, there is a similar observation at the level of attention heads. Therefore, we propose multi-task model profiling to identify these task-specific/task-agnostic layers and heads. As an initial work, we believe that PASTA can motivate future work on attention steering.

---

### Official Review · Reviewer_ppCr · 2023-11-01

**Soundness:** 3 good
**Presentation:** 4 excellent
**Contribution:** 3 good
**Rating:** 6
**Confidence:** 4

**Summary:**

This paper proposes a novel attention steering approach, PASTA, to make the model focus on a subset of the prompt/instruction specified by the user at inference time for large language models. PASTA re-weights the attention scores by reducing the scores of non-highlighted tokens by a factor of $\alpha \sim 0.01$. After normalization, the attention scores of the highlighted tokens are increased. The modified attention matrix is used for computing the projections of the tokens, which are further passed to the feed-forward layers and eventually the next transformer layer. The paper also proposes a multi-task model profiling algorithm to select a subset of the heads in each layer of the model. Empirically, this leads to performance gains over simple attention head-selection heuristics or using all the attention heads. The paper conducts experiments on four synthetic tasks - JSON Formatting, Pronouns changing, Counterfact, and BiasBios using the LLama-7B and GPT-J models. The proposed approach achieves significant improvements over zero-shot, *-marked, ""-marked, and few-shot prompting methods.

**Strengths:**

- The paper is well written and fairly easy to follow.
- The proposed approach is novel and intuitive as increasing the attention scores of highlighted tokens will lead to more contribution in the output projection of tokens during generation.
- The multi-task model profiling to select a certain subset of attention head improving performance on evaluation tasks leads to better generalization even on unseen tasks.
- The empirical results on the four tasks show significant improvements over the standard zero-shot and few-shot prompting baselines.

**Weaknesses:**

- The paper mentions that the proposed approach, PASTA, does not need access to model weights, but I believe that is not correct? Once the attention scores have been modified, you would need access to $W_{v_{h}}$ to compute $H^{(l, h)}$ in equation 1. Furthermore, you'll need the weight matrices of the two feedforward layers $W_{ffn1}$ and $W_{ffn2}$ (assuming non-gated activation) to compute the output of a given transformer block layer, which can be passed to the next layer. Please correct me if I am wrong, but I think that the next layer's output cannot be computed without access to these weights.
- The results are on those tasks only where the highlighted instruction is at the end of the prompt (except for the rephrased JSON Formatting task). What is the general effect of having highlighted instruction at the start or some other place in the prompt? This would give an idea if the proposed approach is general enough.
- Not a weakness, but there's a typo in footnote 3: access $\textbf{to}$ model weights ...

**Questions:**

I have asked most of my questions in the weakness section, but here are some followup questions that I am interested in:

- Beam search sampling and top-k sampling are much better compared to greedy. The baseline performance would definitely improve using these generation methods. How does PASTA compare to the baselines using these sampling approaches?
- *-marked and "" - marked have very low performance. Have the authors tried better markers like <mark> </mark> or <emphasize> </emphasize>. I believe these markers are much more frequent in the training set, and might improve performance.

Also, there are a couple of formatting issues:
- Appendix section A is empty in the submission. Please remove that in the updated version.
- Citations and links are coloured. They should be black for ICLR.

---

> ### Author Response · Authors · 2023-11-21
> **We appreciate the reviewer for the valuable suggestions.**
>
> Thanks for your helpful suggestions. We provide our response to your questions as follows:
>
> **Q1: The weight matrices of two FFN layers and value projects are required to compute the next layers’s output.**
>
> We appreciate the reviewer for raising this question. To further clarify, our method is not designed for black-box models. PASTA requires reading and updating attention scores calculated by attention modules. For model weights or other intermediate outputs, PASTA does not interact with, read, or modify them. The steering operation is strictly limited to the attention mechanism. If the model only provides the attention scores during the forward pass, we still can effectively apply PASTA.
>
> **Q2: What is the general effect of having highlighted instruction at the start or some other place in the prompt?**
>
> We would like to remind the reviewer that both highlighted input spans in BiasBios and CounterFact tasks are not at the end of the prompts. As mentioned in Section 4, we highlight the first sentence for the BiasBios task, which carries the most information for prediction. In the CounterFact task, we highlight new facts, which are in the middle of prompts. Meanwhile, Table 4 already shows the performance of PASTA on the JSON Formatting task when the highlighted instruction is at the beginning of prompts. These results demonstrate that PASTA consistently improves the performance no matter where highlighted input spans are.
>
> In the following table, we further present the PASTA performance for LLAMA-7B when the highlighted instruction is at the start of the Pronouns changing prompts.
>
> | Instruction Position | Method | Acc | All-changed Acc |
> | :-----: | :-----: | :-----: | :-----: |
> | Start of prompts  | Zero-shot | 19.63 | 9.82 |
> | Start of prompts | PASTA | 50.96 | 46.06 |
> | End of prompts | Zero-shot | 71.84 | 66.28 |
> | End of prompts | PASTA | 96.42 | 95.84 |
>
> The results illustrate the sensitivity of model performance to prompt rephrasing – having the instruction at the start for Pronouns changing task degenerates the accuracy of zero-shot prompting to 19.63%. In contrast, PASTA can improve its performance by 31.3%, indicating the effectiveness of PASTA for highlighting instructions at different positions.
>
> **Q3: The baseline performance would definitely improve using these generation methods. How does PASTA compare to the baselines using these sampling approaches?**
>
> To clarify the reviewer’s question, we apply the greedy search for all of the methods including prompting strategies and PASTA to have a fair comparison. Moreover, in the BiasBios and CounterFact tasks, models mainly use the logit of the first generated tokens to make predictions, and hence greedy search should be used. The following table presents the zero-shot performance of LLAMA-7B on the JSON Formatting and Prons. Changing tasks when applying beam search and topk sampling.
>
> | Method | Sampling strategy | JSON F./P. Acc | Prons. Acc/A. Acc |
> | :-----: | :-----: | :-----: | :-----: |
> | Zero-shot | Topk (k=5) | 35.93 / 26.67 | 59.90 / 54.26 |
> | Zero-shot | Topk (k=50) | 19.73 / 10.42 | 53.80 / 46.58 |
> | Zero-shot | Beam search (# beams=4) | 15.81 / 14.95 | 86.95 / 83.73 |
> | PASTA(multi-task) | Beam search (# beams=4) | 83.97 / 71.11 | 98.06 / 97.36 |
> | PASTA(multi-task) | Gready search | 96.64 / 85.09 | 96.42 / 95.48 |
>
> From the results, we can see that PASTA still outperforms baselines when using the beam search or topk sampling strategies.
>
>
> **Q4: \*-marked and "" - marked have very low performance. Have the authors tried better markers like <mark> </mark> or <emphasize> </emphasize>.**
>
> In the following table, we present the performance of these marked prompting baselines on LLAMA-7B across four tasks.
>
> | Method | JSON F./P. Acc | Prons. Acc/A.Acc | BiasBios Acc | CounterFact ES/PS |
> | :-----: | :-----: | :-----: | :-----: | :-----: |
> | <mark></mark>-marked | 9.90/9.32 | 28.99/25.25 | 90.22 | 58.08/49.33 |
> | <emphasize></emphasize>-marked | 8.14/7.66 | 31.83/27.23 | 90.76 | 62.78/ - |
> | PASTA (multi-task) | 96.64/85.09 | 96.42/95.84 | 95.28 | 99.60/99.57 |
>
> **Q5: Response to other questions.**
>
> We appreciate the reviewer for pointing out typos in our paper and other suggestions for the presentation. We will follow these suggestions and make revisions in the next version.

---

> > ### Comment · Reviewer_ppCr · 2023-11-22
> > **Response to Rebuttal**
> >
> > Thank you for providing a detailed response to the review. However, I still believe that access to non-attention weight matrices is required when you are modifying the attention scores. If the attention weights of layer $l - 1$ are changed, the corresponding transformation will also be changed due to the presence of the value matrix in the attention block, and access to the value matrix is required to pass the modified representation to the feedforward layer and consequently, the next $l^{th}$ layer.
> >
> > Given this point and a few other issues raised by the reviewers, I maintain my original rating.

---

### Meta-Review · Area_Chair_QtrE · 2023-12-06

**Metareview:**

This paper introduces PASTA (Post-hoc Attention STeering Approach), which applies attention reweighting at inference time to let the model focus more on user-specified parts of the text. Experiment results show that PASTA significantly improves LLMs’ ability to follow user instructions (such as “output in json format”).

The reviewers generally appreciate the novelty and clarity of the paper. However, a few concerns are raised, including:
1. Modifying attention weights at inference time could potentially hurt generation quality (pBs7 and WBEq). WBEq is particularly concerned with whether the proposed eval is able to capture this. (“As such, an empty generation "" would score 100. This explains partially Figure 3b, where the model's fluency seems to be decreasing steadily as more heads are steered, while the pronoun change accuracy keeps increasing.”)
2. The method would be replaced with instruction-tuning.

For the evaluation issue, the paper uses automatic metrics to capture the (potential drop) of fluency and consistency. For the concern #2, the authors have included additional results on instruction-tuned models showing that the two methods are orthogonal (https://openreview.net/forum?id=xZDWO0oejD&noteId=CKlHR7px18)

**Justification For Why Not Higher Score:**

* The eval metrics has some issues as reviewer WBEq pointed out.
* In practice, trading off between steerability and generation quality is going to be tricky.

**Justification For Why Not Lower Score:**

The approach is interesting and the evaluation is extensive despite its flaws.

---

### Decision · Program_Chairs · 2024-01-16

Accept (poster)